# The Cretaceous physiological adaptation of angiosperms to a declining $pCO_2$ : a modeling approach emulating paleo-traits

Julia Bres[1], Pierre Sepulchre[1], Nicolas Viovy[1], and Nicolas Vuichard[1]

[1]Laboratoire des Sciences du Climat et de l'Environnement, LSCE/IPSL, CEA-CNRS-UVSQ, Université Paris-Saclay, 91191 Gif-sur-Yvette, France

**Correspondence:** Julia Bres (julia.bres@lsce.ipsl.fr)

**Abstract.** The Cretaceous evolution of angiosperm leaves towards higher vein densities enables unprecedented leaf stomatal conductance. Still, simulating and quantifying the impact of such change on plant productivity and transpiration in the specific environmental conditions of the Cretaceous remains challenging. Here, we address this issue by combining a paleo proxy-based model with a full atmosphere-vegetation model that couples stomatal conductance to carbon assimilation. Based on the fossil record, we build and evaluate three consistent proto-angiosperm vegetation parameterizations under two end-members scenarios of $pCO_2$ (280 ppm and 1120 ppm) for the mid-Cretaceous : a reduction of leaf hydraulic or photosynthetic capacity and a combination of both, supported by a likely coevolution of stomatal conductance and photosynthetic biochemistry. Our results suggest that decreasing leaf hydraulic or/and photosynthetic capacities always generate a reduction of transpiration that is predominantly the result of plant productivity variations modulated by light, water availability in the soil, atmospheric evaporative demand and $pCO_2$. The high $pCO_2$ acts as a fertilizer on plant productivity that strengthens plant transpiration and water-use efficiency. However, we show that proto-angiosperm physiology does not allow vegetation to grow under low $pCO_2$ because of a positive feedback between leaf stomatal conductance and leaf area index. Our modeling approach stresses the need to better represent paleovegetation physiological traits. It also confirms the hypothesis of a likely evolution of angiosperms from a state of low leaf hydraulic and photosynthetic capacities at high $pCO_2$ to a state of high leaf hydraulic and photosynthetic capacities linked to leaves with more and more veins together with a more efficient biochemistry at low $pCO_2$.

## 1 Introduction

Vegetation plays a pivotal role in the climate system as it controls water and energy fluxes at the interface between land surfaces and the atmosphere through albedo (Charney et al., 1975; Port et al., 2016; Brovkin et al., 2009), roughness length, and evapotranspiration capabilities (e.g., Bathiany et al., 2010; Betts et al., 1997; Kleidon et al., 2000; Gibbard et al., 2005). Specifically, evapotranspiration, i.e. the sum of soil evaporation, vegetation evaporation and vegetation transpiration fluxes, is a key term in the continental hydrological cycle, and has been shown to control moisture convergence and convection, thereby precipitation patterns in the tropics either in present-day-like configurations (Sun and Barros, 2015; Fraedrich et al., 1999) or for past climates (e.g., Braconnot et al., 1999; Brovkin et al., 2006). It is therefore mandatory that earth system models designed to simulate climate evolution, either in the past or in the future, account for plant traits that alter these fluxes, despite the numerous

challenges involved (Fisher and Koven, 2020). Including vegetation traits in land surface models is a growing field of research (Kattge et al., 2020), but mostly dedicated to better represent present-day vegetation and its response to human-induced climate change (Scheiter et al., 2013; Dury et al., 2018; Peng et al., 2020; Davin and de Noblet-Ducoudré, 2010). The problem has an additional degree of complexity for million-years-old ("deeptime") climates, for which the vegetation physiological traits have been very different from the present. As shown recently for the extinct vegetation of the late Paleozoic, fossil plants provide invaluable information regarding "paleo-traits" that can be in turn included in land surface models (White et al., 2020; Richey et al., 2021).

Another case-study for testing the impact of pivotal changes in physiological traits on the vegetation characteristics is the Cretaceous angiosperm radiation (Boyce and Lee, 2010; Boyce et al., 2010; De Boer et al., 2012). With more than 295,000 species described, angiosperms represent more than 95 % of modern terrestrial vascular plant diversity (Christenhusz and Byng, 2016). One of the most emblematic changes in vegetation through Earth history is their diversification at the expense of gymnosperms during the Cretaceous (Condamine et al., 2020). The macroflora fossil record reveals significant changes in leaf anatomy between the early Cretaceous and the late Cretaceous associated with the rise of angiosperms. Specifically, measurements on fossil leaves show that vein density ($D_v$, see Table 1 for more information about variable abbreviation, definition and unit) increased from 3.1 mm mm$^{-2}$ for the oldest angiosperms leaves from the Aptian-Albian (115 Ma) to 9.8 mm mm$^{-2}$ by the Maastrichtian-Paleocene (66 Ma), while gymnosperms $D_v$ remained centered around 2.4 mm mm$^{-2}$ over the same period (Feild et al., 2011a). Contemporaneous changes in stomatal density (size), hereafter called $D_s$ (S), are observed in the fossil record showing a strong positive (negative) correlation with vein density (De Boer et al., 2012). While veins allow the plants to efficiently transport water from the soil close to the site of transpiration (i.e. stomata), the latter allow the leaves to efficiently exchange carbon and water with the atmosphere. Studies have already shown that $D_v$ is a reliable marker of leaf hydraulic capacity (Brodribb et al., 2007; Brodribb and Feild, 2010; Feild et al., 2011a, b; Boyce et al., 2010; Lee and Boyce, 2010; Boyce et al., 2009; Boyce and Lee, 2010, 2017) and pioneer modeling studies have explored the impact of such a variation of vein density on the transpiration rate in present-day conditions (Boyce et al., 2010; Lee and Boyce, 2010; Boyce and Lee, 2010, 2017). Based on the closed link between vein density, leaf hydraulic and photosynthetic capacities (Brodribb et al., 2007), linear relationships have been inferred between $D_v$ and (i) the transpiration rate on one hand (Boyce et al., 2009; Boyce and Lee, 2017; Boyce et al., 2010), and (ii) the maximum rate of carboxylation ($V_{cmax}$), involved in photosynthesis, on the other hand (Boyce and Lee, 2010; Lee and Boyce, 2010). Sensitivity tests carried out with a land surface model coupled to an atmospheric circulation model showed that angiosperms with modern traits strengthen the current hydrological cycle, ensuring seasonally high levels of transpiration and precipitation rates in the tropics (Lee and Boyce, 2010; Boyce and Lee, 2010, 2017; Boyce et al., 2010).

Paleobotanical data show that several traits from the soil-plant-atmosphere continuum have evolved during the Cretaceous. Specifically, root thickness and wood structure have evolved towards greater hydraulic efficiency (Brundrett, 2002; Wheeler and Baas, 2019). In this paper, we focus on the upper part of this continuum, considering changes in leaf stomatal conductance and leaf photosynthetic capacity. In particular our approach relies on the integration of factors emulating evolution of fossil leaf traits in a land surface model to quantify how transpiration and photosynthesis responded to the angiosperm leaf trait

evolution in the specific environmental conditions of the Cretaceous (different paleogeography and high atmospheric $CO_2$ concentrations). Land surface models used in climate modeling do not explicitly represent traits such as vein densities but rather directly simulate the operational stomatal conductance at the leaf scale ($g_s$) (Farquhar et al., 1980; Ball et al., 1987; Krinner et al., 2005; Yin and Struik, 2009). However, based on relationship obtained from extant and fossil leaves that links maximal anatomic stomatal conductance to $H_2O$ ($g_{anat}^{max}$) to $D_v$ (Brodribb et al., 2007; Brodribb and Feild, 2010), they can be parameterized to represent the changes in stomatal conductance that occured with the angiosperm radiation. Depending on their design, these parameterizations can reflect changes on leaf traits controlling leaf hydraulic capacity, leaf photosynthetic capacity or both. The latter is supported by our knowledge of modern plant processes and in particular of a coevolution between stomatal conductance and $V_{cmax}$ (Franks and Beerling, 2009a; De Boer et al., 2012). Although there is no fossil evidence of changes in leaf photosynthetic capacity, we assume that this covariation is a time-invariant property and that such relationship was the same in the past. Moreover, atmospheric $CO_2$ concentration ($pCO_2$) is known to be one of the drivers that control the stomatal opening and closing at very short-term (Jarvis et al., 1999). Several studies have reported that the $g_s/g_{anat}^{max}$ ratio is relatively stable around 20-40 %, an optimal range within which changes in guard cell turgor pressure are most efficient in controlling quick stomatal conductance changes (Dow et al., 2014; Dow and Bergmann, 2014; McElwain et al., 2016). Therefore, it is likely that $pCO_2$ also drives changes in $g_{anat}^{max}$ on long time scales. Paleo-$CO_2$ reconstructions based on proxies (plant fossils and isotopes) and geochemical models show that, despite a large spread, $pCO_2$ was high throughout the Cretaceous, ranging from 500 ppm to 2000 ppm. Values peaked during the mid-Cretaceous (Cenomanian-Turonian, ca. 95 Ma), then steadily declined towards the Cretaceous-Paleogene boundary (Fletcher et al., 2008; Wang et al., 2014). It is thus important to consider this varying $pCO_2$ when exploring the mechanisms of the angiosperm physiological changes.

Finally, the goal of our study is to evaluate the combined environmental and physiological-induced response of Cretaceous vegetation to a low $D_v$ representative of proto-angiosperms. Our experimental design considers both reducing leaf hydraulic or/and photosynthetic capacities and applying different $pCO_2$ scenarios in a coupled atmosphere-vegetation model. The modeled $g_s$ allows to account for long-term plant adaptation to their changing environment. We evaluate the physiological response of the vegetation to low hydraulic and photosynthetic capacities at the leaf scale, then at the global scale. It leads us to evaluate water and carbon exchanges between the land-surface and the atmosphere.

**Table 1.** List of main variables used in the models and their units.

| Variable | Definition | Units |
|---|---|---|
| A | Net carbon assimilation | $molCO_2$ m$^{-2}$[leaf] s$^{-1}$ |
| A1 | Empirical constant | - |
| $\alpha$ | Factor describing changes in leaf hydraulic capacity | - |
| B1 | Empirical constant | kPa$^{-1}$ |
| $\beta$ | Factor describing changes in leaf photosynthetic capacity | - |
| $C_i$ | Intercellular $CO_2$ partial pressure | bar |
| $C_i^*$ | $C_i$-based $CO_2$ compensation point in the absence of $R_d$ | bar |
| $\Delta\Psi_{leaf}$ | Leaf water potential gradient | MPa |
| $D_v$ | Vein density | mm mm$^{-2}$[leaf] |
| fcpl | Parameter for describing the effect of external forcing on $g_s$ | - |
| $g_0$ | Residual operating stomatal conductance when irradiance approaches zero | mol m$^{-2}$[leaf] s$^{-1}$ |
| $g_c$ | Operating canopy stomatal conductance | mol m$^{-2}$[ground] s$^{-1}$ |
| $g_{anat}^{max}$ | Maximal anatomic stomatal conductance | mol m$^{-2}$[leaf] s$^{-1}$ |
| GPP | Gross Primary Productivity | kgC m$^{-2}$[ground] y$^{-1}$ |
| $g_s$ | Operating leaf stomatal conductance | mol m$^{-2}$[leaf] s$^{-1}$ |
| J | Rate of e$^-$ transport | mol e$^-$ m$^{-2}$[leaf] s$^{-1}$ |
| $K_c$ | Michaelis–Menten constant of Rubisco for $CO_2$ | bar |
| $K_o$ | Michaelis–Menten constant of Rubisco for $O_2$ | bar |
| LAI | Leaf Area Index | m$^2$[leaf] m$^{-2}$[ground] |
| O | Oxygen partial pressure | bar |
| $\rho$ | Air density | kg m$^{-3}$ |
| q | Specific humidity of the surrounding air | kg kg$^{-1}$ |
| $q_s^*$ | Saturation specific humidity of the evaporating surface | kg kg$^{-1}$ |
| $r_a$ | Aerodynamic resistance | s m$^{-1}$ |
| $r_c$ | Canopy resistance | s m$^{-1}$ |
| $R_d$ | Daily respiration (respiratory $CO_2$ release other than by photorespiration) | $molCO_2$ m$^{-2}$[leaf] s$^{-1}$ |
| Tr | Transpiration rate | mm d$^{-1}$ |
| VPD | Leaf-to-air vapor pressure deficit | kPa or MPa |
| $V_c$ | Rubisco enzyme activity | $molCO_2$ m$^{-2}$[leaf] s$^{-1}$ |
| $V_{cmax}$ | Maximum rate of Rubisco activity-limited carboxylation | $molCO_2$ m$^{-2}$[leaf] s$^{-1}$ |
| $V_j$ | Regeneration rate of the Rubisco | $molCO_2$ m$^{-2}$[leaf] s$^{-1}$ |
| $V_{jmax}$ | Maximum value of J under saturated light | mol e$^-$ m$^{-2}$[leaf] s$^{-1}$ |
| $w_{soil}^{lim}$ | Soil moisture stress for transpiration | - |
| y | Distance from vein terminals to epidermis | $\mu$m |

## 2 Methods

### 2.1 IPSL atmosphere-land surface model

In this study, we use LMDZ and ORCHIDEE versions embedded in the IPSL-CM5A2 model (Sepulchre et al., 2020). LMDZ is the atmosphere general circulation model of the Institut Pierre-Simon Laplace earth system model. LMDZ couples a dynamical core solving the primitive equations of conservation with a set of physical parameterizations of the radiative and convective schemes. The spatial resolution of LMDZ is the regular 3.75 x 1.875° longitude-latitude grid. LMDZ has 39 vertical levels to describe vertical processes (Hourdin et al., 2013). ORCHIDEE is the land surface component of the IPSL earth system model (Krinner et al., 2005). When coupled with LMDZ, it has the same horizontal resolution, and a global 4-meter soil water depth represented with a two-layer bucket model, that is adequate to simulate evapotranspiration fluxes consistent with the data (Guimberteau et al., 2014). ORCHIDEE simulates water and energy exchanges (Rosnay and Polcher, 1998; Ducoudré et al., 1993), and the key processes of the terrestrial carbon cycle such as photosynthesis, carbon allocation, or soil organic matter decomposition for thirteen Plant Functional Types (PFTs). Each PFT gathers a set of plant species with similar functional characteristics that will be translated into a common set of parameter values. Photosynthesis, respiration, water and energy exchanges are computed every 30 minutes, while carbon-related slow processes (leaf phenology and carbon allocation) are computed on a daily basis (Krinner et al., 2005; Sitch et al., 2003).

#### 2.1.1 Stomatal conductance in ORCHIDEE

Stomatal conductance is computed per canopy layer, with layer depth increasing exponentially from top to bottom of the canopy. This feature enables to account for the response to light radiation extinction through the canopy. Leaf stomatal conductance is coupled to photosynthesis based on original formulations from Ball et al. (1987) and Leuning et al. (1995). Within the model used in ORCHIDEE (Yin and Struik, 2009), leaf operational stomatal conductance to $H_2O$ ($g_s$ in mol m$^{-2}$[leaf] s$^{-1}$) depends on the net carbon assimilation (A in molCO$_2$ m$^{-2}$[leaf] s$^{-1}$), the daily respiration ($R_d$ in molCO$_2$ m$^{-2}$[leaf] s$^{-1}$), the intercellular $CO_2$ partial pressure ($C_i$ in bar), the $C_i$-based $CO_2$ compensation point in the absence of $R_d$ ($C_i^*$ in bar) and the residual stomatal conductance to $H_2O$ ($g_0$ in mol m$^{-2}$[leaf] s$^{-1}$), to account for a non-zero conductance when the carbon assimilation is zero (Farquhar et al., 1980; Ball et al., 1987; Yin and Struik, 2009). Finally, the leaf operational stomatal conductance to $H_2O$ is modulated by a factor fcpl, describing the strength of the coupling between A and $g_s$, which is function of the leaf-to-air vapor pressure deficit (kPa) and that we will further name "leaf hydraulic capacity" :

$$g_s = g_0 + \frac{A + R_d}{C_i - C_i^\star} * fcpl \tag{1}$$

The intercellular $CO_2$ partial pressure is dependant of the leaf boundary layer conductance and the ambient air $CO_2$ partial pressure. The coupling factor relies on two empirical constants A1 (unitless) and B1 (kPa$^{-1}$) :

$$fcpl = \frac{1}{\frac{1}{(A1 - B1*VPD)} - 1} \tag{2}$$

The photosynthesis scheme proposed by Yin and Struik (2009) solves analytically the net assimilation, the stomatal conductance and the leaf intercellular $CO_2$ concentration. The semi-empirical formalism of $g_s$ allows to account for both the structural conductance, linked to the morphologic and physiologic traits of leaves (e.g. $D_v$, $D_s$ and S) developed on the long-term plant evolution history, and the dynamical conductance, related to the short-term stomatal opening and closing depending on the environment. A is limited by either the Rubisco enzyme (RuBP) activity ($V_c$) or the electron transport ($V_j$):

$$A = min(V_c, V_j) \tag{3}$$

$V_c$ (molCO$_2$ m$^{-2}$[leaf] s$^{-1}$) represents the ability of the RuBP to fix $CO_2$ and initiate the Calvin cycle. It relies on $V_{cmax}$ (molCO$_2$ m$^{-2}$[leaf] s$^{-1}$), $K_c$ and $K_o$ the Michaelis-Menten constants of Rubisco for $CO_2$ and $O_2$ respectively (bar) and O the oxygen partial pressure (bar):

$$V_c = \left[ \frac{(C_i - C_i^\star)V_{cmax}}{C_i + K_c\left(1 + \frac{O}{K_O}\right)} - R_d \right] * \left[1 - w_{soil}^{lim}\right] \tag{4}$$

$V_{cmax}$ is parameterized for each PFT as a function of leaf age (Krinner et al., 2005). $V_j$ (molCO$_2$ m$^{-2}$[leaf] s$^{-1}$) represents the regeneration rate of the RuBP. The latter depends on the radiation and the electron transport :

$$V_j = \left[ \frac{(C_i - C_i^\star)J}{4C_i + 8C_i^\star} - R_d \right] * \left[1 - w_{soil}^{lim}\right] \tag{5}$$

Where J (mol e$^-$ m$^{-2}$[leaf] s$^{-1}$) is the actual rate of electron transport (Yin and Struik, 2009), dependent on irradiance and $V_{jmax}$, the maximum rate of electron transport at saturating light (mol e$^-$ m$^{-2}$[leaf] s$^{-1}$). $V_{jmax}$ is directly proportional to $V_{cmax}$ with the ratio $V_{cmax}/V_{jmax}$ varying with surface monthly temperature (Kattge and Knorr, 2007). Then, when discussing the changes in $V_{cmax}$, we consider the proportional changes in $V_{jmax}$, in the following. A increases with $V_{cmax}$ and pCO$_2$, until to reach a plateau where light and temperature become the limiting factors. Both $V_c$ and $V_j$ are a function of the soil moisture stress for transpiration $w_{soil}^{lim}$, that ultimately controls photosynthesis. This factor depends on both the soil moisture and the root profile given per PFT, trees having deeper roots than grasses. $w_{soil}^{lim}$ is unitless and ranges between 0 for fully moist soil and 1 for maximal water stress. The transpiration rate Tr (mm d$^{-1}$) is a function of the potential evapotranspiration, that is, the atmospheric evaporative demand, modulated by a serie of resistances :

$$Tr = \frac{1}{r_c + r_a}\rho(q_s^\star - q) \tag{6}$$

where $r_a$ and $r_c$ are the aerodynamic and canopy resistances respectively (s m$^{-1}$), $\rho$ is the air density (kg m$^{-3}$), $q_s^*$ is the saturation specific humidity of the evaporating surface (kg kg$^{-1}$), function of the temperature, and q is the specific humidity of the surrounding air (kg kg$^{-1}$). Calculation of the transpiration requires the integration of $g_s$ at the canopy level which is $g_c$ (molH$_2$O m$^{-2}$[ground] s$^{-1}$), that is over the leaf area index (LAI) and expressed as :

$$g_c = \int_{l=0}^{l=LAI} g_s(l)\,\mathrm{d}l$$
$$g_c = \frac{1}{r_c} \tag{7}$$

## 2.1.2  Summary of dynamical coupling between stomatal conductance and photosynthesis in a given environment

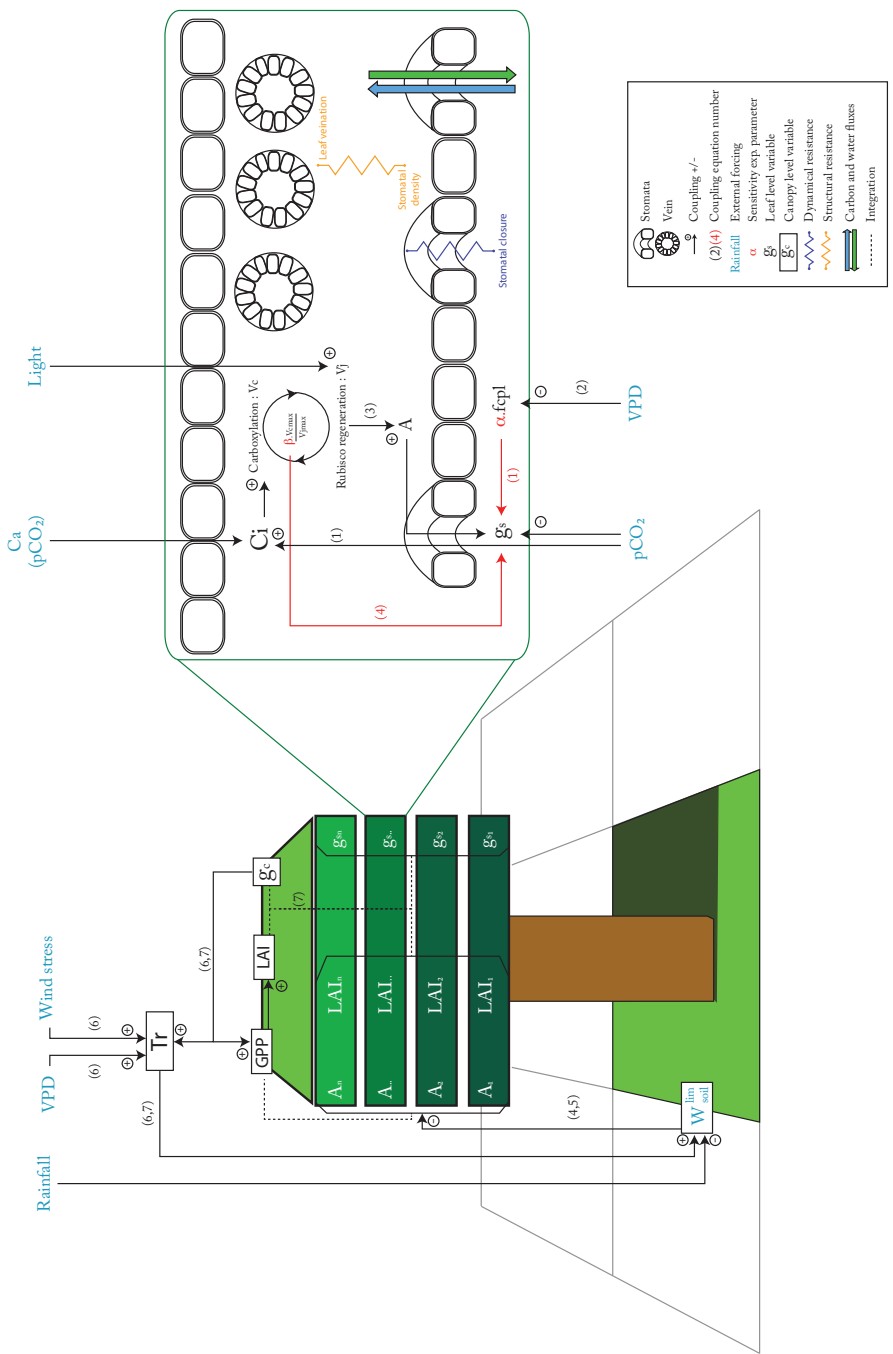

**Figure 1.** Schematized coupling between stomatal conductance and photosynthesis from the leaf to the canopy scale in the ORCHIDEE model.

At the leaf level, the structural resistance (Fig. 1, in orange) represents traits (veins and stomata) that evolve at long time scales. The structural resistance drives the maximum theoretical conductance of water ($g_{anat}^{max}$). The dynamical resistance (Fig. 1, in dark blue) controls the stomatal closure on a very short time scale, depending to the external forcing : $pCO_2$, light, vapor pressure deficit and soil moisture stress for transpiration (Fig. 1, in light blue). However, the structural resistance defines the upper boundary condition that constrains the dynamical resistance. The coupling factor (fcpl) controls the sensitivity of stomatal conductance to the surrounding environment : a low fcpl implies that $g_s$ is less sensitive to $C_i$, A and the vapor pressure deficit than a high fcpl. Plants can also slightly adapt their $V_{cmax}$ and $V_{jmax}$, in absence of nitrogen stress, on seasonal timescale and in a greater range on evolutionary long time scales. The ORCHIDEE version used in this study does not include the nitrogen cycle, as we do not have good constraints regarding the C:N ratio of Cretaceous vegetation and soils. The end limit of the assimilation process is the available energy from light that controls the quantity of RuBP that can be regenerated ($V_j$ in Eq. (5)). At high $pCO_2$, $C_i$ is maintained at a sufficient level to feed the carboxylation rate ($V_c$) and then, for a sufficient $V_{cmax}$, light becomes the main limiting factor even for a small conductance (Eq. (3) to (5)). So, the structural resistance is not a limiting factor and plants will close the stomata to limit the water loss. Likewise, $V_c$ rate increases, with both $C_i$ and $V_{cmax}$, and the level of $V_{cmax}$ for which $V_j$ becomes the main limiting factor decreases (Eq. (3)). Because nitrogen acquisition and RuBP protein maintenance consume a lot of energy, plants tend to optimize the $V_{cmax}$ and $V_{jmax}$ to reach the point where $V_c$ and $V_j$ are colimiting (Maire et al., 2012; Stocker et al., 2020). On the contrary, at low $pCO_2$ structural conductance (that drives the maximum conductance) can become a limiting factor since, even with full stomata opening, it can limit $C_i$ under the level to feed the carboxylation rate. $V_{cmax}$ and $V_{jmax}$ should also be increased to maintain a sufficient level of carboxylation ($V_c$) (lower efficiency of the oxidation/carboxylation ratio).

At the canopy level (Fig. 1), the canopy conductance $g_c$ depends on both leaf level conductance $g_s$ and LAI. As LAI is a function of plant productivity, there is an additional positive feedback: a change in $g_s$ will impact GPP and then LAI in the same direction, which will amplify the initial effect of $g_s$ on $g_c$. Another external forcing is the soil water availability ($w_{soil}^{lim}$). In the model used, water stress acts directly on $V_{cmax}$ and $V_{jmax}$, as supported by previous studies (Keenan et al., 2009, 2010; Egea et al., 2011). A low soil water content will induce a water stress limiting the $V_{cmax}$ and $V_{jmax}$ that will indirectly also reduce $g_s$. Hence, arid ecosystems will be less sensitive to structural change in $g_s$ than ecosystems without a large hydric stress. On the contrary, they will be affected in the same way for direct changes on $V_{cmax}$ and $V_{jmax}$.

## 2.2 Fossil evidence of increasing angiosperm leaf hydraulic and photosynthetic capacities

Vein density as well as stomatal size and density are both used to reconstruct past variations of $g_{anat}^{max}$ (Franks and Beerling, 2009a, b; Brodribb et al., 2007; Brodribb and Feild, 2010; De Boer et al., 2012; Franks and Farquhar, 2001). Here, we have chosen to account for $D_v$ changes rather than $D_s$ and S. Indeed, using $D_v$ is a good proxy to constrain $g_{anat}^{max}$ since $D_v$ and $D_s$ are correlated and that observed relationship between $D_v$ and $g_{anat}^{max}$ gives the highest correlation coefficient (Feild et al., 2011b). Vein densities from angiosperm fossils published in Feild et al. (2011b) record a 2 to 5-fold increase in angiosperm $D_v$ during the Cretaceous (Fig. 2a), compared to early angiosperms and non-angiosperms (Table S2).

$D_v$ allows to reconstruct the maximal water that can flow through the stomata $g_{anat}^{max}$ (mol m$^{-2}$[leaf] s$^{-1}$) using the relationship developed by Brodribb et al. (2007) and Brodribb and Feild (2010) :

$$g_{anat}^{max} = \frac{12760}{VPD} \Delta\Psi_{leaf} \left( \frac{\pi}{2} \sqrt{\frac{650^2}{D_v^2} + y^2} \right)^{-1.27} \tag{8}$$

Where VPD is the leaf-to-air vapor pressure deficit (MPa), $\Delta\Psi_{leaf}$ is the leaf water potential gradient (MPa), $D_v$ is the vein density (mm mm$^{-2}$[leaf]) and y is the distance from vein terminals to epidermis ($\mu$m). The past VPD and $\Delta\Psi_{leaf}$ are set to 0.002 MPa and 0.4 MPa respectively, values that are typical for temperate-tropical environments (Brodribb and Feild, 2010). An estimation of y is 70-130 $\mu$m, producing a span of predicted values encompassing the likely morphological variability of leaves (Brodribb and Feild, 2010).

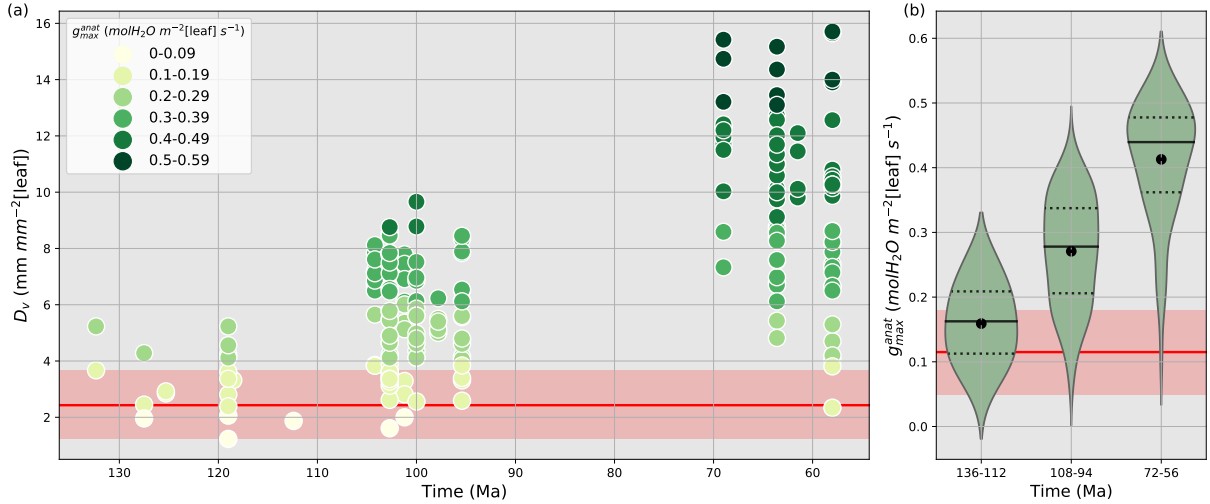

**Figure 2.** (a) Time evolution of fossil vein densities (mm mm$^{-2}$[leaf], y-axis) adapted from Feild et al. (2011a) and the corresponding maximal stomatal conductance to H$_2$O (mol m$^{-2}$[leaf] s$^{-1}$, filled circles) calculated by the anatomic relationship (Eq. 8) developed by Brodribb et al. (2007) and Brodribb and Feild (2010). (b) Subdivision of maximal stomatal conductance to H$_2$O (mol m$^{-2}$[leaf] s$^{-1}$) by time periods. The black solid lines are the angiosperm medians, the black dotted lines are the first and third quartiles, and the black points are the angiosperm mean values for each time period. Red lines and areas show the gymnosperm mean values and the gymnosperm mean $\pm$ one standard deviation area respectively, over the Cretaceous.

A lower value for y indicates a lower stomata-to-vein distance and a higher $g_{anat}^{max}$. The highest increase in $g_{anat}^{max}$ is obtained for the largest variation of $D_v$ over time combined with the smallest y. Conversely, the lowest increase in $g_{anat}^{max}$ is obtained for the smallest variation of the highest $D_v$ values over time, combined with the highest y (Table S2). From the early to the late Cretaceous, the 2 to 5-fold increase in angiosperm $D_v$ corresponds to a 3 to 5-fold increase in $g_{anat}^{max}$ (Fig. 2b and Table S2). Thus, fossil $D_v$ provides an estimation of the increase in $g_{anat}^{max}$ over time. Our land surface model does not explicitly represent

vegetation traits nor $g_{anat}^{max}$ but only $g_s$, the operational stomatal conductance. However, based on the strong relationship between $g_{anat}^{max}$ and $g_s$ (Dow et al., 2014; Dow and Bergmann, 2014), one assumes that variations of $g_s$ due to the long-term evolution

of leaf hydraulic and photosynthetic capacities together with that of environmental factors such as pCO$_2$ would reflect into proportional changes on g$_{anat}^{max}$.

## 2.3 Experimental setup

To assess the impact of angiosperm leaf evolution on transpiration and photosynthesis, we performed climate model simulations while varying leaf hydraulic or photosynthetic capacity, or both, for two contrasted pCO$_2$ levels (Table 2). This is not a direct functional trait-based modeling approach, but we mimic paleo-traits through constraints on the maximum operating stomatal conductance g$_s$ with different factors. The model continental boundary conditions include mid-Cretaceous (115 Ma) paleogeography reconstructions from Sewall et al. (2007). Global vegetation distribution is set by establishing a first-order estimate of correspondence between biomes, inferred from paleobotanical data (Sewall et al., 2007), and PFTs (Table S1). Figure 3 shows only the dominant PFT, but several PFTs can be present in a pixel. For instance, regions indicated as C3 grass are savannas with a fraction of trees (Table S1). We prescribe PFT distribution rather than activate the dynamical vegetation model to constrain our interpretations of the sensitivity experiments and avoid additional feedback linked to vegetation dynamics. This configuration allows the carbon cycle to be activated and the LAI to be prognostically calculated as a function of dynamic carbon allocation (Krinner et al., 2005).

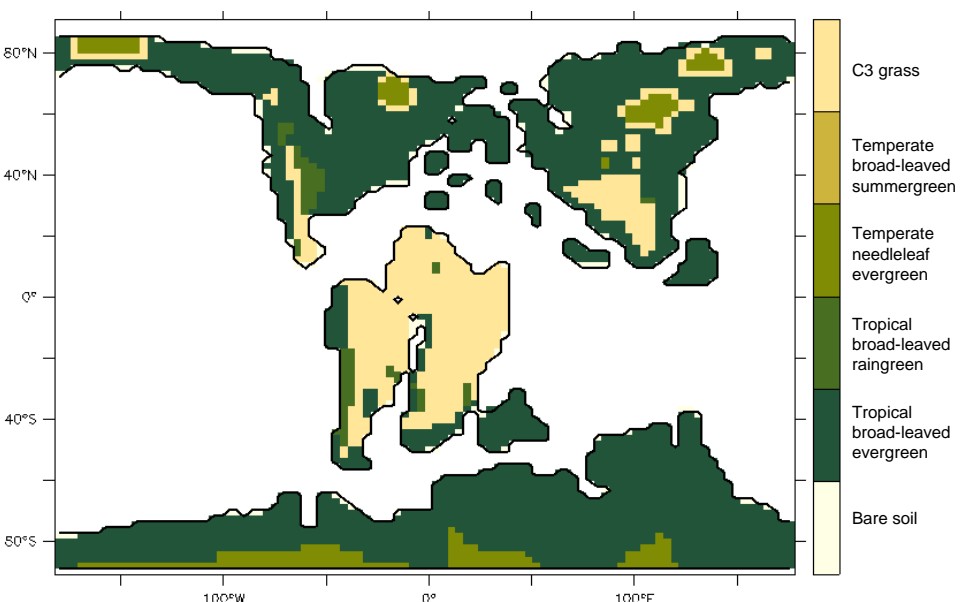

**Figure 3.** Vegetation distribution prescribed in the Aptian simulation configuration : map showing the dominant ORCHIDEE PFT for each grid-cell. For all experiments, this distribution is fixed through time. The prescribed fraction of bare soil is given Fig. S1.

Although scattered paleo soil texture data do exist for the Cretaceous, sensitivity studies carried on present-day have shown a rather weak sensitivity of the terrestrial water budget to soil texture in the ORCHIDEE model (Tafasca et al., 2020). Thus, we set this parameter worldwide to an averaged value corresponding to loam in Zobler classification (Zobler, 1986) : 0.39 % silt,

0.43 % sand and 0.18 % clay. Soil color determines the bare soil albedo and is set globally at 0.16 (Wilson and Henderson-

Sellers, 1985). We set the solar radiation at 99 % of the current value (Gough, 1981) and the orbital parameters as today
(Laskar et al., 2004). The ocean is prescribed through sea-surface temperatures obtained from an unpublished Aptian fully-
coupled simulation obtained with the ESM IPSL-CM5A2 (Sepulchre et al., 2020). To assess the impact of angiosperm trait
evolution on transpiration and photosynthesis, we consider (i) two different factors applied to angiosperm-PFTs only, and (ii)
one external forcing ($pCO_2$), that alter the stomatal conductance in our land-surface model (Table 2).

**Table 2.** Definition of the experiments

| Name | Factors representing the fraction of current leaf capacities for angiosperms PFTs | | $pCO_2$ (ppm) | Description |
|---|---|---|---|---|
| | Hydraulic capacity ($\alpha$) | Photosynthetic capacity ($\beta$) | | |
| ANGIO(1120) | 1 | 1 | 1120 | Cretaceous world with the high leaf hydraulic and photosynthetic capacities of modern angiosperms |
| NOANGIOh(1120) | $1/5$ | 1 | 1120 | Cretaceous world without the high leaf hydraulic capacity of modern angiosperms |
| NOANGIOp(1120) | 1 | $1/5$ | 1120 | Cretaceous world without the high leaf photosynthetic capacity of modern angiosperms |
| NOANGIOhp(1120) | $\sqrt{1/5}$ | $\sqrt{1/5}$ | 1120 | Cretaceous world without the high leaf hydraulic and photosynthetic capacities of modern angiosperms |
| ANGIO(280) | 1 | 1 | 280 | Cretaceous world with the high leaf hydraulic and photosynthetic capacities of modern angiosperms, under preindustrial $pCO_2$ |
| NOANGIOh(280) | $1/5$ | 1 | 280 | Cretaceous world without the high leaf hydraulic capacity of modern angiosperms, under preindustrial $pCO_2$ |
| NOANGIOp(280) | 1 | $1/5$ | 280 | Cretaceous world without the high leaf photosynthetic capacity of modern angiosperms, under preindustrial $pCO_2$ |
| NOANGIOhp(280) | $\sqrt{1/5}$ | $\sqrt{1/5}$ | 280 | Cretaceous world without the high leaf hydraulic and photosynthetic capacities of modern angiosperms, under preindustrial $pCO_2$ |

Change in leaf hydraulic capacity is considered via a change in the coupling factor (fcpl) that describes the slope of the relationship between $g_s$ and the dynamical driving parameters (namely A, $pCO_2$ and vapor pressure deficit). Because fcpl is related to the structural conductance, the change in $g_{anat}^{max}$ can then be empirically represented by a factor $\alpha$, modulating fcpl in the $g_s$ formulation (Eq. (1)). Changes in leaf photosynthetic capacity are taken into account via changes in $V_{cmax}$ (Eq. (4)). To this end, we use a factor $\beta$ that is applied to both $V_{cmax}$ and $V_{jmax}$ (i.e $V_{cmax}$/$V_{jmax}$ ratio kept constant, Eq. (3) to (5)). We carry

out three different factorial experiments using the upper value of changes in $g_{anat}^{max}$ induced from $D_v$ proxy as depicted earlier. (i) $\alpha$ is set to $^1/_5$ while $\beta$ is 1, accounting for a direct decrease of angiosperm leaf hydraulic capacity by a factor of 5 (NOANGIOh, Table 2). (ii) $\beta$ is set to $^1/_5$ while $\alpha$ is 1, corresponding to a direct decrease of angiosperm leaf photosynthetic capacity by a factor of 5 (NOANGIOp, Table 2). This sensitivity experiment is supported by the coupling between stomatal conductance and assimilation in the model and by several studies suggesting to mimic proto-angiosperm capacities by decreasing modern

$V_{cmax}$ by a factor of 5 (Boyce and Lee, 2010; Lee and Boyce, 2010). Since leaf hydraulic and photosynthetic capacities are likely to have co-evolved over the Cretaceous (Franks and Beerling, 2009a; De Boer et al., 2012), (iii) we set $\alpha$ and $\beta$ to $\sqrt{1/5}$, i.e. with half the forcing simultaneously applied to the leaf hydraulic and photosynthetic capacities (NOANGIOhp, Table 2). These three sensitivity tests represent three likely Cretaceous worlds with proto-angiosperm types. We perform the reference control experiment where the factors $\alpha$ and $\beta$ are set to 1 in order to keep the modern leaf hydraulic and photosynthetic capacity

of extant angiosperms as the reference ones (ANGIO, Table 2). Since $pCO_2$ simultaneously impacts leaf hydraulic and photosynthetic traits evolution on the long-term (Franks and Beerling, 2009a), we repeat this set of experiments for two different extreme $pCO_2$ forcings. At 1120 ppm, we refer to the mid-Cretaceous (115 Ma) estimates, contemporary with the beginning of the angiosperm radiation. At 280 ppm, we make a sensitivity test to assess the response of proto-angiosperm vegetation to a preindustrial-like $pCO_2$. Regarding the average $g_s$ value over a long time period, we assume that the $g_s$/$g_{anat}^{max}$ ratio is a constant

at long-time scales because of plants structural adaptation to their environment (Dow et al., 2014; Dow and Bergmann, 2014). We consider that all the changes in the model operational $g_s$, arising from the above-mentioned proto-angiosperm parameterizations, reflect that of the maximal anatomic stomatal conductance. The experiments were run for 60 years, a sufficient time to balance gross primary productivity and evapotranspiration. We analyse and show the averaged last 10 years of simulations.

## 3   Results

### 3.1   Leaf operating stomatal conductance

The leaf stomatal conductance to $H_2O$ ($g_s$, mol m$^{-2}$[leaf] s$^{-1}$) is the global mean of annual daylight leaf stomatal conductance of the top of canopy leaves (where exposure to sunlight is maximal). Figure 4 shows that leaf stomatal conductance decreases as we sequentially reduce leaf hydraulic (NOANGIOh) or photosynthetic (NOANGIOp) capacities, or both (NOANGIOhp), to mimic the proto-angiosperm world. Reducing the high leaf hydraulic and/or photosynthetic capacities from modern angiosperm

leads to a 3-fold drop of leaf stomatal conductance on average, a factor in the lowest bracket of the range expected from $g_{anat}^{max}$ inferred from the fossil record (Fig. 2b). Within every PFT group, $g_s$ decrease is similar amongst sensitivity experiments and validates our chosen combinations of values for $\alpha$ and $\beta$. The maximal decrease in $g_s$ is simulated for the tropical broad-leaved

evergreen and the C3 grass PFTs which also depict the largest stomatal conductance in the reference ANGIO simulations. Leaf
hydraulic and photosynthetic capacities have not been altered for the temperate needleleaf evergreen PFT, as they correspond
to gymnosperms. Thus, gymnosperm $g_s$ was expected to be constant for each of the pCO$_2$ group of experiments. However, $g_s$
slightly decreases in the sensitivity experiments for this PFT, suggesting a feedback from the atmosphere-vegetation coupling
that ultimately altered soil moisture stress and stomatal conductance. Indeed, soil moisture stress for gymnosperm transpiration
increases for all the simulations with the proto-angioperm vegetation prescribed (Fig. S2) because of less precipitations (Fig.
S3).


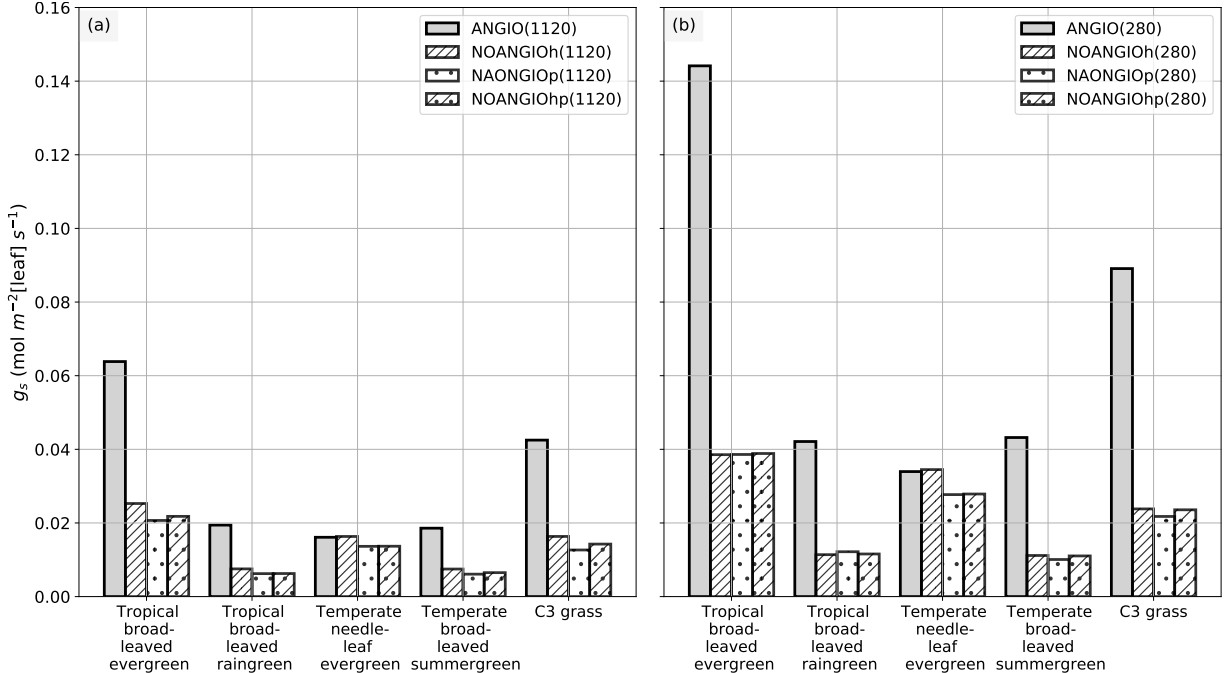

**Figure 4.** Global scale leaf operating stomatal conductance to H$_2$O $g_s$ (mol m$^{-2}$[leaf] s$^{-1}$) per PFT : (a) ANGIO(1120) (gray bar), NOAN-
GIOh(1120) (diagonal hatched bar), NOANGIOp(1120) (dotted bar), and NOANGIOhp(1120) (diagonal hatched and dotted bar); (b) AN-
GIO(280) (gray bar), NOANGIOh(280) (diagonal hatched bar), NOANGIOp(280) (dotted bar), and NOANGIOhp(280) (diagonal hatched
and dotted bar). The leaf stomatal conductance to H$_2$O is the global mean of annual daylight leaf stomatal conductance of the top of canopy
leaves.

Simulations forced with "Cretaceous-like" pCO$_2$ (1120 ppm) show systematically lower $g_s$ values than simulations with pCO$_2$
set at preindustrial (280 ppm). This illustrates that the model well reproduces the expected closure of stomata as a response
to high pCO$_2$, in direct link with Eq. (4) and (5). This decrease in $g_s$ with increasing pCO$_2$ is the highest (52 % on aver-
age) for experiments with the modern-like leaf hydraulic and photosynthetic capacities (ANGIO(1120) versus ANGIO(280)).
For these experiments, tropical broad-leaved evergreen PFT reacts to the high pCO$_2$ with $g_s$ values dropping from 0.14 mol
m$^{-2}$[leaf] s$^{-1}$ at 280 ppm to 0.06 mol m$^{-2}$[leaf] s$^{-1}$ at 1120 ppm, while that of C3 grasses drops from 0.09 to 0.04 mol

$m^{-2}$[leaf] $s^{-1}$. At 1120 ppm, as the leaf stomatal conductance of ANGIO runs is already strongly reduced compared to that at 280 ppm, the relative decrease in $g_s$ when modern leaf hydraulic and photosynthetic capacities are reduced is also less important : - 66 % for tropical broad-leaved evergreen at 1120 ppm to be compared to -73 % for the same PFT at 280 ppm (Fig. 4).


## 3.2 Leaf area index and vegetation cover

Annual mean leaf area index (LAI) provides an indication of the vegetation ability to grow given the prescribed boundary conditions, i.e. Cretaceous paleogeography, varying $pCO_2$ and interactions with the atmosphere. It is the weighted average of leaf area index of each PFT per surface of ground of a grid-cell (i.e. surface averaged across all PFTs including bare soil, written ground$_{CELL}$). For the modern-like vegetation (ANGIO), figures 5a and b show that the highest LAI values are simulated for regions dominated by tropical broad-leaved evergreen, that are set in mid-to-high latitudes in the specific Cretaceous configuration (Fig. 3). LAI is also slightly higher at 1120 ppm than at 280 ppm, as a response to $CO_2$ fertilization effect on photosynthesis that increases biomass production (Fig. 1). The vegetation response to the prescribed perturbations of fcpl and $V_{cmax}$ strongly differs depending on the $pCO_2$. At high $pCO_2$, LAI is unchanged with the sole perturbation of the leaf hydraulic capacity (fcpl*$1/5$, NOANGIOh, Fig. 5c) meaning that even with a reduced $g_s$ at the leaf level, the high $pCO_2$ allows for the fertilization effect to drive biomass growing ($C_i$ in Eq. (1)). Conversely, $V_{cmax}$ perturbation (NOANGIOp, Fig. 5e) induces a strong decrease in the vegetation cover of all angiosperm PFTs, that is replaced by bare soil (Fig. S4), with LAI dropping to values ranging from 1 to 4 $m^2$[leaf] $m^{-2}$[ground$_{CELL}$] compared to LAI ranging from 1 to 8 $m^2$[leaf] $m^{-2}$[ground$_{CELL}$] for ANGIO(1120). Lastly, when both fcpl and $V_{cmax}$ perturbations are combined (fcpl*$\sqrt{1/5}$ and $V_{cmax}$*$\sqrt{1/5}$, NOANGIOhp, Fig. 5g), LAI is mostly unchanged, suggesting that in a high $pCO_2$ context, slightly reduced leaf hydraulic and photosynthetic capacities does not affect actual assimilation. At high $pCO_2$, the impact of reducing $V_{cmax}$ on photosynthesis is limited because the limiting factor is likely $V_j$ (and then light) rather than $V_c$ (and $C_i$) (Eq. (3)). At low $pCO_2$, LAI depicts a much stronger response to perturbations. Reducing the leaf hydraulic capacity leads to a global drop of the vegetation cover, with grass-dominated regions averaging 1 $m^2$[leaf] $m^{-2}$[ground] and tree-dominated regions barely reaching 5 $m^2$[leaf] $m^{-2}$[ground] except in the very southern latitudes, where no significant change is simulated (Fig. 5d). The strongest signal occurs with the reduction of leaf photosynthetic capacity through $V_{cmax}$ (NOANGIOp, Fig. 5f). Apart from the unchanged needleleaf PFT, the modified vegetation cannot grow in NOANGIOp experiment at 280 ppm, as shown by LAI values close to zero for every angiosperm PFTs. Simulated LAI in NOANGIOhp shows that at a lower $V_{cmax}$ associated with a lower fcpl, tropical PFTs also globally collapse, although vegetation is maintained at high latitude (Fig. 5h and S4h). At 280 ppm, intercellular $CO_2$ partial pressure (Eq. (1) to (5)) becomes more limiting than the solar energy whatever the vegetation prescribed. The latitudinal gradient of LAI reduction testifies to another limiting factor to be integrated, which plays a role at low latitudes but not at high latitudes. Figure S5 shows that the soil moisture stress for transpiration (Fig. 1, Eq. (4) and (5)) is larger in the tropics to mid-latitudes and near zero at high latitudes. As expected, water in the soil available for transpiration is thus a limiting factor of plant growth in the tropics and mid-latitudes at low $pCO_2$.

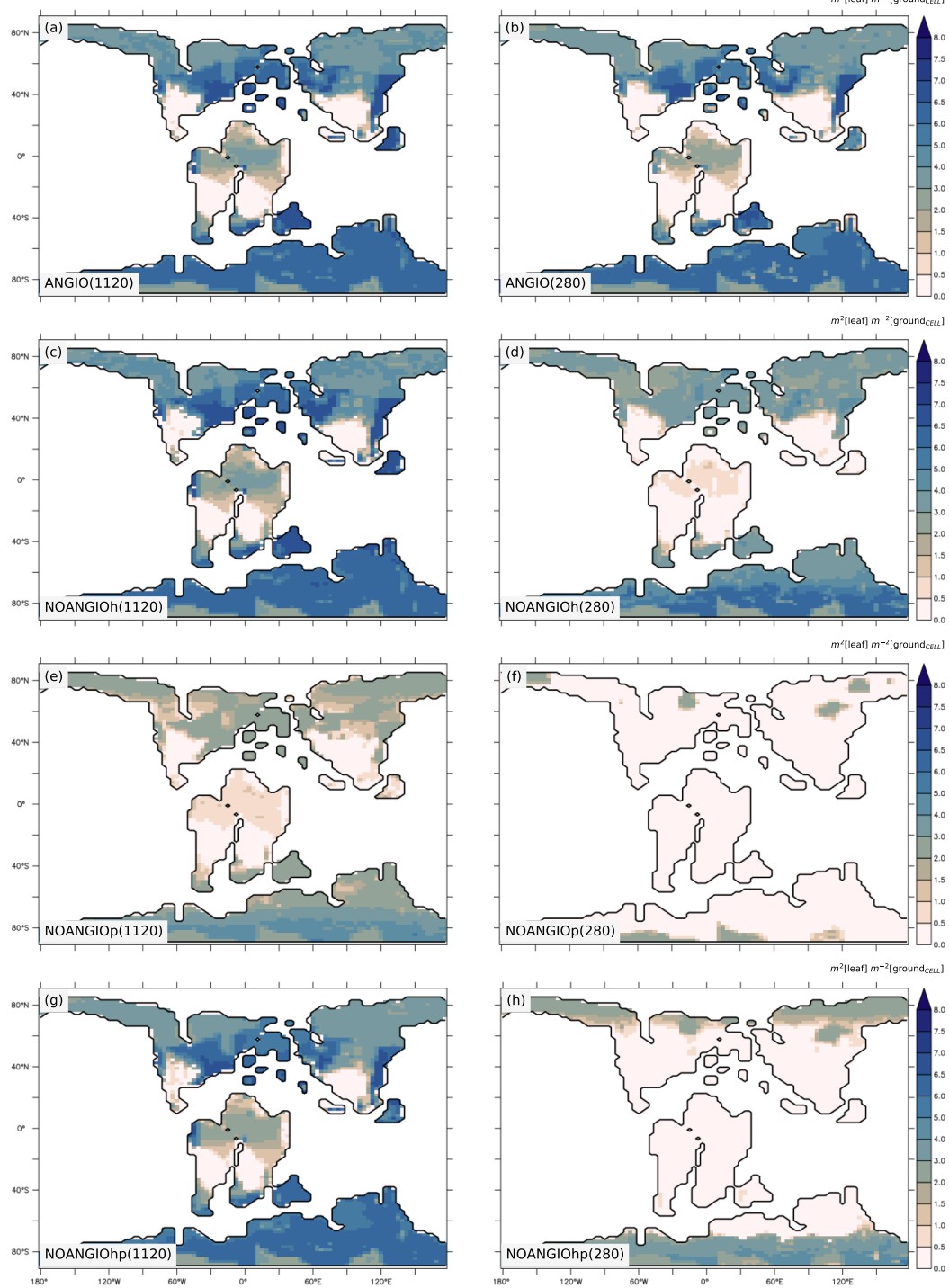

**Figure 5.** Annual mean LAI (m$^2$[leaf] m$^{-2}$[ground$_{CELL}$]) for (a) ANGIO(1120), (b) ANGIO(280), (c) NOANGIOh(1120), (d) NOAN-GIOh(280), (e) NOANGIOp(1120), (f) NOANGIOp(280), (g) NOANGIOhp(1120) and (h) NOANGIOhp(280). The annual mean LAI is the weighted average of leaf area index of each PFT per surface of ground of a grid-cell (i.e. surface averaged across all PFTs including bare soil, that is written ground$_{CELL}$ in the text).

## 3.3 Canopy operating stomatal conductance

Albeit the fossil record and empirical models can help to infer the foliar maximal stomatal conductance to $H_2O$, the land surface model is required to explore the stomatal conductance to $H_2O$ at the canopy level that is given by Eq. (7) : $g_c$ is the integral of $g_s$ over the entire canopy and is inversely proportional to the canopy resistance $r_c$. The canopy stomatal conductance is the global mean of annual daylight canopy stomatal conductance and is weighted by surface unit of ground of a grid-cell where each PFT is found (i.e. written $ground_{PFT}$). $g_c$ is the combined result of changes in $g_s$ arising from the forcing factors $\alpha$ and $\beta$, and changes in vegetation leaf area, i.e., both have an influence that can either be reinforcing or counterbalancing. As expected, figures 4 and 6 show significant differences between $g_s$ and $g_c$ in respect to the $pCO_2$ prescribed: while $g_s$ decreases by about a factor of 3 for each proto-angiosperm sensitivity experiment compared to the reference (Fig. 4), the decrease in $g_c$ is not equal through experiments (Fig. 6). The dramatic drop in $g_c$ when reducing the high leaf photosynthetic capacity (NOANGIOp) is well explained by the significant LAI decrease (Fig. 5e and f). Besides, the effect of $pCO_2$ is more complex at the canopy scale. Indeed, $g_c$ is still reduced in the perturbed experiments, but is not systematically lower at 1120 ppm than at 280 ppm (Fig. 6) as opposed to $g_s$ (Fig. 4).

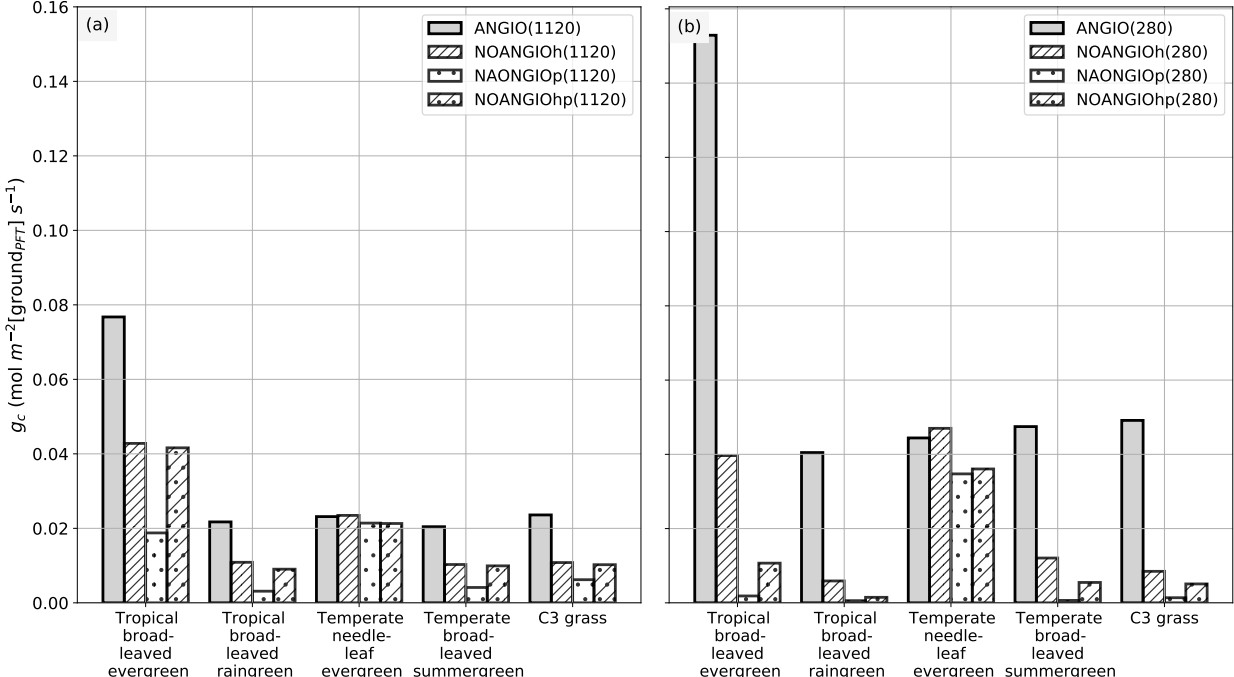

**Figure 6.** Global scale canopy operating stomatal conductance to $H_2O$, $g_c$ (mol m$^{-2}$[ground$_{PFT}$] s$^{-1}$) per PFT : (a) ANGIO(1120) (gray bar), NOANGIOh(1120) (diagonal hatched bar), NOANGIOp(1120) (dotted bar), and NOANGIOhp(1120) (diagonal hatched and dotted bar); (b) ANGIO(280) (gray bar), NOANGIOh(280) (diagonal hatched bar), NOANGIOp(280) (dotted bar), and NOANGIOhp(280) (diagonal hatched and dotted bar). The canopy stomatal conductance to $H_2O$ is the daylight average over the year and is weighted by the surface of ground of a grid-cell where each PFT is found (written ground$_{PFT}$ in the text).

NOANGIOh simulations do not show a change in $g_c$ between 280 ppm and 1120 ppm. A decrease in fcpl by a factor of 5 implies a reduction of the LAI at 280 ppm (Fig. 5d) because plants cannot assimilate enough carbon to sustain their biomass ($C_i$ in Eq. (1) to (5)) whereas it is not the case at 1120 ppm (Fig. 5c). Indeed, at the canopy scale, at low $pCO_2$, higher $g_s$ just makes up the LAI loss. Conversely, at high $pCO_2$, lower $g_s$ is compensated by the higher LAI, ultimately making angiosperm $g_c$ being almost identical for the two $pCO_2$ scenarios. In the NOANGIOp and NOANGIOhp experiments, $g_c$ is higher at 1120 ppm than at 280 ppm. By increasing the carbon assimilation during photosynthesis, high $pCO_2$ allows for higher vegetation cover (Fig. 5e and g) and higher canopy stomatal conductance whereas low $pCO_2$ together with low leaf photosynthetic capacity imply a collapse of the vegetation (Fig. 5f and h).

LAI changes described earlier modulate the relationship between $g_s$ and $g_c$ (Fig. 1). At low $pCO_2$, the LAI decrease between the ANGIO and any of the three perturbed experiments (Fig. 5d, f and h) strengthens the initial decrease in $g_s$ (positive feedback, Fig. 4b and 6b). Mean $g_s$ decreases by 69 %, 73 % and 69 % while mean $g_c$ decreases by 72 %, 97 % and 91 % respectively for NOANGIOh, NOANGIOp and NOANGIOhp compared to ANGIO (Fig. 4b and 6b). Indeed, decreasing $V_{cmax}$ under low $pCO_2$ implies a reduction of the plant capacity to assimilate carbon (Eq. (4)) and directly impacts the GPP (Fig. S6f and h) and then the LAI at the canopy scale (Fig. 5f and h). However, decreasing fcpl and thus $g_s$ reduce the $CO_2$ concentration at the chloroplast level ($C_i$ in Eq. (1)) and have only an indirect effect on GPP (Fig. S6d) and thus on LAI (Fig. 5d). At high $pCO_2$, the LAI is almost sustained for NOANGIOh and NOANGIOhp compared to ANGIO (Fig. 5a, c and g) because the assimilation remains high when $C_i$ is not the limiting factor (Fig. S6a, c and g). The latter lessens the initial decrease in $g_s$ (Fig. 6a) on $g_c$ (Fig. 4a). Nevertheless, NOANGIOp(1120) experiment shows a much lower $g_c$ than the two previous experiments because of the direct impact of decreasing $V_{cmax}$ on the LAI (Fig. 5e). Therefore, comparing $g_c$ of perturbed simulations to that of the reference allows us to account for the structural conductance linked to plant trait evolution at the canopy level.

### 3.4 Transpiration rate

As shown in Eq. (6), transpiration rate is controlled by (i) the atmospheric evaporative demand (Fig. S7), that depends on air vapor pressure deficit, the surface temperature (Fig. S8) as well as the aerodynamic resistance, and (ii) the capacity of plants to transpire, driven by the canopy conductance (Fig. S9). For both $pCO_2$ cases, experiments with modern-like vegetation depict the highest transpiration rates in the tropics and the mid-latitudes, where they reach up to 2.5 and 3 mm day$^{-1}$ at 1120 and 280 ppm, respectively (Fig. 7a and b). These regions have optimal conditions in water availability in the soil (Fig. S5) and light, resulting in higher canopy stomatal conductance (Fig. S9a and b) together with a high atmospheric evaporative demand (Fig. S7a and b) as a response to high temperatures (Fig. S8a and b). Transpiration is also slightly strengthened (+ 0.2 mm day$^{-1}$) at low $pCO_2$ compared to high $pCO_2$ (Fig. 7a and b) as a consequence of higher $g_c$ (Fig. 4 and 6). Parameterizing the vegetation without the modern angiosperm leaf hydraulic and photosynthetic capacities systematically leads to lower transpiration rates (Fig. 7).

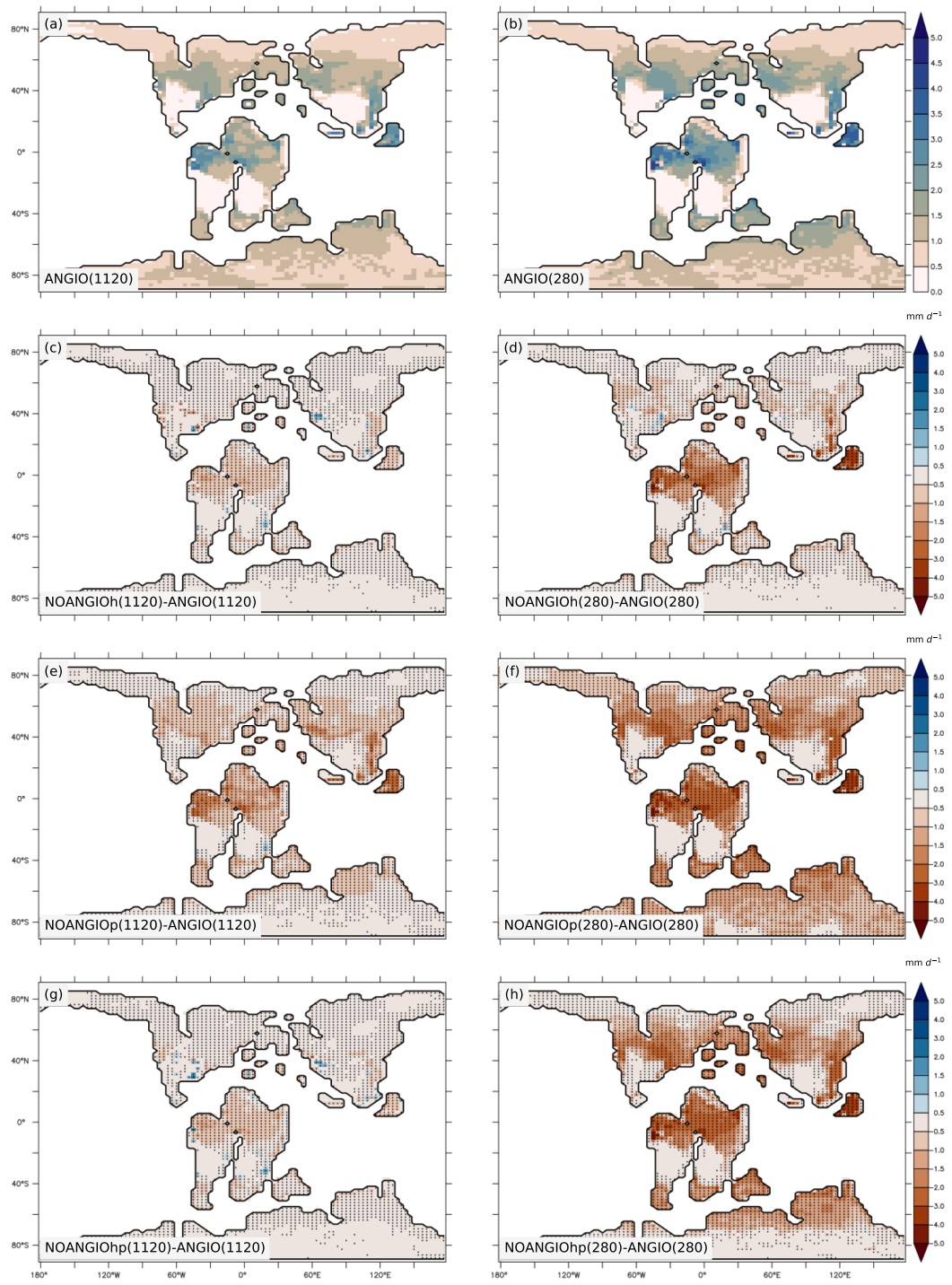

**Figure 7.** Annual mean transpiration rate (mm d$^{-1}$) for (a) ANGIO(1120) and (b) ANGIO(280), anomalies of annual mean transpiration rate (mm d$^{-1}$) for (c) NOANGIOh(1120) vs ANGIO(1120), (d) NOANGIOh(280) vs ANGIO(280), (e) NOANGIOp(1120) vs ANGIO(1120), (f) NOANGIOp(280) vs ANGIO(280), (g) NOANGIOhp(1120) vs ANGIO(1120) and (h) NOANGIOhp(280) vs ANGIO(280). The t-test 95 % confidence level anomalies are given by dots.

Overall, in line with changes depicted for LAI and $g_c$, transpiration rates react stronger in a 280 ppm world than in a 1120 ppm world to decreasing leaf hydraulic and photosynthetic capacities. At 280 ppm, NOANGIOh shows a decrease of 0.6 mm d$^{-1}$ (- 44 %) in transpiration compared to ANGIO, especially over equatorial Gondwana and paleo Southeast Asia, whereas the decrease is limited to 0.3 mm d$^{-1}$ (- 24 %) at 1120 ppm (Fig. 7c and d) as a response to $g_c$ changes described earlier

345 (Fig. 6). Transpiration also significantly drops when leaf photosynthetic capacity alone is reduced (NOANGIOp, Fig. 7e and f). At 1120 ppm, transpiration drops by 0.5 mm d$^{-1}$ (-53 %) (Fig. 7e). The signal is stronger at 280 ppm, where a complete collapse of transpiration is simulated (Fig. 7f). This latter result is a direct consequence of the LAI collapse described earlier (Fig. 5e and f). Combining reduction in leaf photosynthetic and hydraulic capacities (NOANGIOhp) leads to little decreases in transpiration rate at 1120 ppm (Fig. 7g), comparable to NOANGIOh (Fig. 7c), because the high pCO$_2$ prevents the decrease

350 in carbon assimilation (Fig. 5g) and canopy stomatal conductance (Fig. 6). Conversely, at low pCO$_2$, the limitation of C$_i$ (Eq. (1), (4) and (5)) implies a collapse of LAI (Fig. 5h) and $g_c$ (Fig. 6) over the tropics and mid-latitudes, and transpiration is near zero for NOANGIOhp, with a decrease of 1 mm d$^{-1}$ (- 81 %) compared to ANGIO (Fig. 7h).

Transpiration anomalies result from variations in the atmospheric evaporative demand (Fig. S7) combined with that of the canopy stomatal conductance (Fig. S9), arising from $g_s$ and LAI changes which are modulated by the soil moisture stress for

355 transpiration and light. Transpiration anomalies are located over the paleo-tropics (Fig. 7c and d) for NOANGIOh compared to ANGIO experiments because decreasing leaf hydraulic capacity by a factor of 5 acts mainly on wet regions where plants efficiently transpire in the ANGIO world (Fig. 1 and Eq. (1) to (5)), while arid belt regions are less sensitive to any change in $g_c$. In theses regions, transpiration is already constrained by water shortage, so the upper stomatal conductance limit is not the limiting factor. However, mid-latitude do not show transpiration anomalies because the decrease in $g_c$ (Fig. 6, S9c and

360 d) is compensated by the increase in atmospheric evaporative demand (Fig. S7c and d) that strengthens transpiration rate. In contrast, decreasing leaf photosynthetic capacity (NOANGIOp) acts more globally on terrestrial plants and explains the widespread transpiration anomalies (Fig. 7e and f), always modulated by the atmospheric evaporative demand (Fig. S7e and f). At 1120 ppm, NOANGIOhp shows the same pattern as NOANGIOh (Fig. 7c and g) because plants are more sensitive to reduction in fcpl than to reduction in V$_{cmax}$ (C$_i$ is not limiting) while the decrease in transpiration is more extended at 280 ppm (Fig.

365 7h) because it arises mainly from the decrease of V$_{cmax}$ at low pCO$_2$ that drives carbon assimilation to decline (Eq. (3) and (4)).

## 3.5 Water use efficiency

The annual mean water use efficiency (WUE) is the ratio of annual mean GPP to annual mean transpiration over each grid point (gC kgH$_2$O$^{-1}$). It pictures the capacity of vegetation to maximise carbon uptake while minimizing water loss and thus is

370 a good indicator of plant adaptation to their environment. For the modern-like vegetation (ANGIO), maximal values of WUE are found in the mid to high-latitudes where optimal conditions of water and light (Fig. 8a and b, Fig. S5) are available for the vegetation. WUE increases with pCO$_2$ : while it barely exceeds 10 gC kgH$_2$O$^{-1}$ at 280 ppm, it reaches more than 15 gC kgH$_2$O$^{-1}$ at 1120 ppm. This is explained by the CO$_2$ fertilization effect generated by the high pCO$_2$ on C$_i$ (Eq. (3) to (5)) that

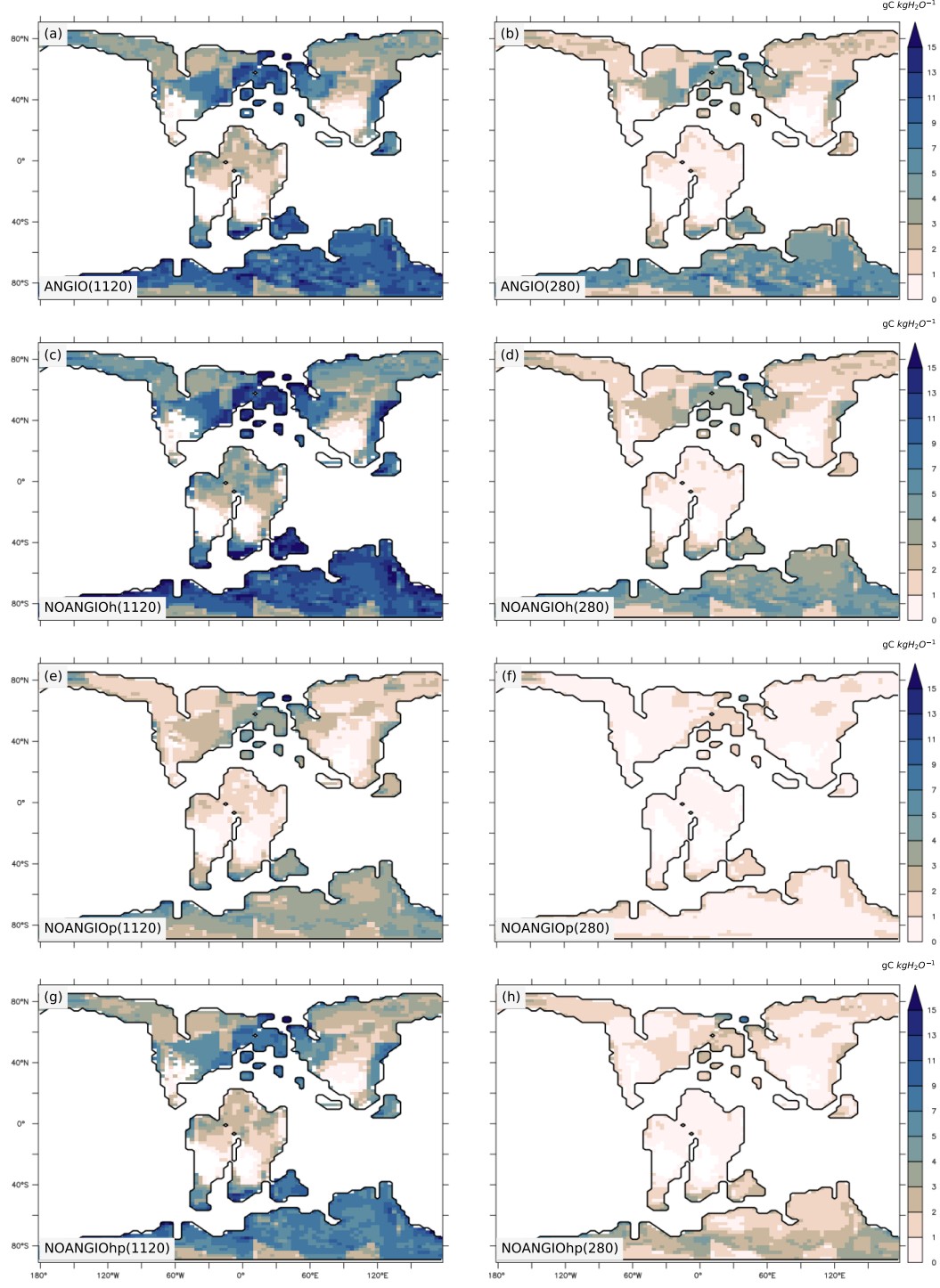

**Figure 8.** Annual mean WUE (gC kgH$_2$O$^{-1}$) for a) ANGIO(1120), b) ANGIO(280), c) NOANGIOh(1120), d) NOANGIOh(280), e) NOAN-GIOp(1120), f) NOANGIOp(280), g) NOANGIOhp(1120) and h) NOANGIOhp(280).

strengthens plant productivity. At 1120 ppm, the highest value of WUE is found for NOANGIOh, with a 1.7 gC kgH$_2$O$^{-1}$
(+ 30 %, Fig. S10c) increase compared to ANGIO (Fig. 8a and c). Indeed, reducing the leaf hydraulic capacity does not
imply a change in GPP which remains similar to that of ANGIO(1120) (Fig. S6a and c) but rather generates a slight decrease in
transpiration rate (Fig. 7c), explaining the increase of WUE. Therefore, plants with lower leaf hydraulic capacity than today are
better adapted to the high pCO$_2$ environment. For NOANGIOhp(1120), the WUE is partly reduced compared to ANGIO(1120)
but stays at a high level (Fig. 8a and g). It results from the larger decrease in GPP than in transpiration rate (Fig. S6g and 7g),
as GPP is more sensitive to the V$_{cmax}$ reduction (Eq. (4)) than transpiration is to the fcpl reduction (Eq. (1) and (6)). In
contrast, WUE is largely degraded in NOANGIOp(1120) compared to ANGIO(1120) (Fig. 8a and e). Both GPP (Fig. S6a and
e) and transpiration rate (Fig. 7a and e) significantly drop when reducing the leaf photosynthetic capacity (NOANGIOp) at
high pCO$_2$. However, V$_{cmax}$ acts directly on the carbon assimilation (Eq. (3) and (4)) while it is more indirect on the stomatal
conductance, thus implying that the reduction of V$_{cmax}$ has a larger effect on GPP than on transpiration rate. In contrast to
experiments at high pCO$_2$, ANGIO(280) gives the highest WUE at low pCO$_2$. NOANGIOh(280) depicts a lower WUE than
ANGIO(280) (Fig. 8b and d) which demonstrates that plants with lower leaf hydraulic capacity than today are less adapted to
the low pCO$_2$ environment. At low pCO$_2$, the low C$_i$ decreases GPP (Eq. (4), Fig. S6b and d) while it increases g$_s$ (Eq. (1), Fig.
4), modulating the transpiration decrease (Fig. 7d). However, at low pCO$_2$, WUE collapses to very low values for NOANGIOp
and NOANGIOhp (Fig. 8f and h), that is driven by the large decrease in GPP when combining the low C$_i$ to the reduction of
V$_{cmax}$ (Fig. S6f and h). Once the leaf photosynthetic capacity is decreased, changes in GPP are the main contributor to the
changes in WUE whatever the pCO$_2$ level prescribed.

## 4   Discussion

### 4.1   How to better account for the proto-angiosperm conductance traits in land surface models?

Fossil maximal anatomic stomatal conductance has been widely used to estimate the maximum of water flow through the
stomata before and after the angiosperm radiation. Still, determining how a 5-time increase in maximal anatomic stomatal
conductance translates into actual flux at the top of the canopy is challenging. We show that the complete response of the
vegetation to evolving physiological and morphological traits is modulated by environmental factors such as pCO$_2$, light,
vapor pressure deficit and water availability in the soil (Fig. 1).

The simplest representation of proto-angiosperm vegetation is to account for the lower maximal anatomic stomatal conductance
by a factor of 5, consistent with the fossil records, directly by applying this factor to fcpl in the calculation of the leaf stomatal
conductance (Eq. (1)). Our results show that a lower fcpl of angiosperms at high pCO$_2$ does not change plant photosynthesis
(Fig. 5c and S6c) but slightly decreases transpiration (Fig. 7c), driving WUE to increase (Fig. 8c) compared to the modern
angiosperm prescription. Hence, a lower maximal stomatal conductance at high pCO$_2$ appears as an advantage compared
to modern angiosperms because of a better optimization of carbon uptake over water loss. At low pCO$_2$, both transpiration
(Fig. 7d) and photosynthesis are decreased because a reduced LAI entails a reduction of canopy stomatal conductance, which
strengthens the initial reduction of leaf stomatal conductance compared to the modern vegetation. Despite the absence of a

direct proxy for fossil plant maximum $V_{cmax}$, several studies have suggested to mimic the proto-angiosperm capacities by decreasing modern $V_{cmax}$ (Boyce and Lee, 2010; Lee and Boyce, 2010), rather than fcpl, by a factor of 5. This approach is supported by our knowledge of modern plant processes that stomatal conductance interacts with assimilation in order to optimize carbon gain against water loss (Bonan, 2015) and is made possible by the coupling between leaf stomatal conductance and leaf photosynthetic capacity in our land-surface model (Farquhar et al., 1980; Ball et al., 1987; Krinner et al., 2005; Yin and Struik, 2009). However, applying this method leads vegetation cover to decrease at high $pCO_2$ (Fig. 5e and S6e) or even collapse at low $pCO_2$ (Fig. 5f and S6f), which is not recorded in the fossil record. As a consequence, transpiration rates (Fig. 7e and f) and WUE (Fig. 8e and f) significantly decrease at high $pCO_2$ and are zero at low $pCO_2$ compared to the modern vegetation. Hence, taking into account stomatal conductance reduction only through $V_{cmax}$ reduction does not appear to be adequate. However, several studies suggested a decrease of $V_{cmax}$ with increasing $pCO_2$ (Ainsworth and Rogers, 2007) driven by (i) a coevolution between stomatal conductance and $V_{cmax}$ through time (Franks and Beerling, 2009a; De Boer et al., 2012) and (ii) the photosynthesis coordination theory that states that plants optimize $V_{cmax}$ to be near the co-limitation between carboxylation rate and RuBP regeneration (Maire et al., 2012; Stocker et al., 2020). This theory has been recently improved considering also the cost related to stomatal conductance (Stocker et al., 2020). This $V_{cmax}$ limitation is related to the high energetic (and then respiration) cost needed to maintain a high level of Rubisco (acquisition of nitrogen). Rather than the two extreme cases that decrease leaf hydraulic or photosynthetic capacity of angiosperms by a factor of 5, we consider a covariation of fcpl and $V_{cmax}$, by applying half the forcing given by the fossil records (i.e. $\sqrt{1/5}$) to fcpl directly and the other half to $V_{cmax}$. Experimental studies on extant plant types (Lin et al., 2015) have shown differences in water-use strategy between modern angiosperm trees and gymnosperm trees. They argue that modern angiosperm trees have 2 times higher stomatal conductance sensitivity response to driving factors than gymnosperm trees, showing that our choice for fcpl*$\sqrt{1/5}$ together with $V_{cmax}$*$\sqrt{1/5}$ seems the most realistic. When applying this factor jointly to fcpl and $V_{cmax}$ at a high $pCO_2$, our results suggest that vegetation is barely impacted, with a slight reduction of GPP (Fig. S6g), that remains sufficient to sustain LAI values close to those of the control scenario (Fig. 5g), and a relatively high WUE (Fig. 8g) compared to the modern vegetation (Fig. 5a and 8a). Conversely at low $pCO_2$, LAI collapses (Fig. 5h) and GPP, transpiration and WUE reach zero (Fig. S6h, 7h and 8h) as a response to carbon assimilation drop.

At 280 ppm, sensitivity tests with a low leaf photosynthetic capacity or together with a low leaf hydraulic capacity can be considered as extrema where plants cannot grow (Fig. 5f and h, S6f and h) whereas experiment with modern trait vegetation corresponds to a maximum in plant productivity and transpiration (Fig. 5b, 7b and 8b). At 1120 ppm, two sets of forcing factors simulate a sustainable paleovegetation productivity : lowered leaf hydraulic capacity only or lowered leaf hydraulic and photosynthetic capacities together (Fig. 5c and g, S6c and g). However, lowering only the leaf hydraulic capacity while keeping the high $V_{cmax}$ as in the modern vegetation is probably not realistic. Indeed, under high $pCO_2$, because of increasing limitation of Rubisco regeneration, the actual photosynthesis can be maintained with a smaller $V_{cmax}$. Sustaining a given level of $V_{cmax}$ is associated with a respiration cost, for nitrogen acquisition and protein maintenance. So, to optimize its carbon gain, we expect the plant to also have a lower $V_{cmax}$. Although our model does not represent the nitrogen cycle, we infer that this supplementary cost would lead to a decrease in productivity when simulating the proto-angiosperm vegetation with a modern

leaf photosynthetic capacity together with a low leaf hydraulic capacity. Taking into account the differentiated response of vegetation to a declining $pCO_2$, we suggest that simultaneously decreasing fcpl and $V_{cmax}$ by a factor of $\sqrt{1/5}$ is the most realistic representation of proto-angiosperm vegetation. The latter is consistent with a previous study of Franks and Beerling (2009a) suggesting that past fluctuations in $pCO_2$ acted as a forcing on both $V_{cmax}$ and stomatal conductance.

Finally, our study confirms the hypothesis that paleovegetation has evolved from a likely state with a relatively low stomatal conductance (Fig. 4a) and $V_{cmax}$ under high $pCO_2$ that allowed nonetheless to sustain a high productivity (Fig. 5g, 7g and 8g) to a state of high stomatal conductance (Fig. 4b) and $V_{cmax}$ as found in the modern vegetation, in order to maintain high productivity under low $pCO_2$ (Fig. 5b, 7b and 8b). The study also shows that paleovegetation characteristics can be better represented by parameterizing models fully describing the coupling between stomatal conductance and plant productivity from leaf to the canopy scale. Furthermore, we show that decreasing leaf hydraulic and/or photosynthetic capacities does not coincide with a decrease of the leaf operational stomatal conductance to the same extent. Indeed, accounting for a decrease by a factor of 5, given by the maximal bound of the range expected from the maximal anatomic stomatal conductance, leaf stomatal conductance is only 3-fold lower than the reference. This difference is due to the fact that plants never operate at the level of the maximal anatomic stomatal conductance because of light and water limitation effects on photosynthesis. While some previous studies directly decrease the transpiration rate by the factor of vein density changes from the fossil record (Boyce et al., 2009; Boyce and Lee, 2017; Boyce et al., 2010), we suggest to explicitly represent changes in leaf hydraulic and photosynthetic capacities. Nevertheless, when embedding parallel changes on $pCO_2$, leaf stomatal conductance of plants with reduced leaf hydraulic and photosynthesis capacities at high $pCO_2$ is nearly 6-time lower than the reference one at low $pCO_2$. It confirms that accounting for $pCO_2$ changes is mandatory to model the evolution of angiosperm leaf traits.

## 4.2    Do the high leaf hydraulic and photosynthetic capacities of angiosperms provide a selective advantage compared to the other plants under decreasing $pCO_2$?

Our work relies upon the assumption that stomata aperture maximizes carbon gain while minimizing water loss (Bonan, 2015). Both photosynthesis and stomatal conductance to $H_2O$ are sensitive to environmental variables such as light, $pCO_2$ and water availability in the soil (Fig. 1). As $pCO_2$ has varied a lot through the Cretaceous (Fletcher et al., 2008; Wang et al., 2014), plants had to adjust their stomatal conductance. Our study suggests that having a high stomatal conductance, which means a large vein density and a high $V_{cmax}$, provides little advantage compared to a low stomatal conductance under high $pCO_2$. In contrast, having a high stomatal conductance under low $pCO_2$ may confer a competitive advantage over plants with limited stomatal conductance to assimilate carbon. But this higher stomatal conductance should be linked to an increase in $V_{cmax}$ to sustain growth. In that sense, fossil records provide evidence of plant traits evolution during the angiosperm radiation. Fossils of early angiosperms show that vein density was as low as that of other plant types (Feild et al., 2011b) because having a high vein density under high $pCO_2$ was not necessary for the plants to grow. Then, the $pCO_2$ likely declined (Fletcher et al., 2008; Wang et al., 2014). For this time, our results support the hypothesis that angiosperms evolved towards leaves with increasing vein density combined with a more efficient biochemistry that allowed them to have an increasing stomatal conductance and photosynthetic capacity to counteract the effect of $pCO_2$ decrease on carbon assimilation. Among other factors, this

evolution of physiological leaf traits has given a competitive advantage to angiosperms compared to gymnosperms dominating the vegetation of the period that enabled them to colonize almost all the terrestrial ecosystems. Our results are consistent with those of Franks and Beerling (2009a), which have shown that WUE co-evolves positively with variations in $pCO_2$ over the Phanerozoic : periods with high $pCO_2$ enhanced GPP while simultaneously allowing a reduction of transpirational water losses due to reduced stomatal conductance. They also show that even after the evolution of angiosperm leaf morphology and biochemistry, WUE is estimated to have been at its lowest level since the Carboniferous. Our model consistently represents the range of WUE deducted by Franks and Beerling (2009a) under different $pCO_2$ : between 5 and 9 gC kgH$_2$O$^{-1}$ (Fig. 8a and b). Moreover, there is a likely co-adaptation of stomatal traits and leaf venation which implies a better optimisation of carbon gain against water loss (De Boer et al., 2012). Progressively, under decreasing $pCO_2$, angiosperms with high stomatal density and low stomatal size (Franks and Beerling, 2009a, b) likely invested increasingly more energy in building more and more veins to sustain the higher stomatal conductance and then carbon assimilation, while other plants did not. The innovation of angiosperms in dense water transport networks could have become a necessity to support higher stomatal conductance and prevent plant desiccation (De Boer et al., 2012).

## 4.3 What are the limitations of our modeling choices?

As a first step toward understanding the impact of trait evolution on the leaf hydraulic and photosynthetic capacities of angiosperms, we have chosen to simulate the Aptian (115 Ma) because this time period corresponds to the first step of increasing vein density found in the fossil record (Feild et al., 2011a). To get an exhaustive view of the angiosperm evolution, future studies will benefit from considering similar experiments with boundary conditions set several million years before and after the Aptian. Specifically, exploring cold and warm extremes of the Cretaceous, such as the Cenomanian-Turonian (95 Ma) and the Maastrichtian (70 Ma) would be valuable, as climate and $pCO_2$ have been shown to vary a lot during these periods (Ladant and Donnadieu, 2016).

The vegetation map we used (Fig. 3) results from two efforts of (i) compilation and spatialization of the Aptian paleobotanical records (Sewall et al., 2007) and (ii) conversion of the fossil data into plant functional types combination (Table S1). Each of these two steps include uncertainties that can propagate into our results, but can be hardly quantified. We acknowledge that the prescribed vegetation cover, especially in the tropics, can potentially alter the radiative balance and the hydrological cycle (e.g., Port et al., 2016). It is however unlikely that the Aptian vegetation cover would be very different from the one provided by Sewall et al. (2007), given the compilation effort made for this reconstruction. Further studies could still circumvent this potential issue by running a full dynamical vegetation model, i.e. by allowing PFTs to spatially settle in regions where the simulated climate is the most appropriate.

Recent studies on past vegetation transitions indicate that differences in transpiration rates also arise from the ratio of carbon over nitrogen (White et al., 2020; Richey et al., 2021). As mentioned earlier, the ORCHIDEE version we used does not explicitly represent the nitrogen cycle, preventing us from considering the additional cost to maintain high leaf photosynthetic capacity with lower leaf hydraulic capacity. The recent developments of a new version of ORCHIDEE that does include the nitrogen cycle (Vuichard et al., 2019), based on previous developments (Zaehle and Friend, 2010; Zaehle et al., 2010), will help

to account for this process, provided that good constraints can be obtained regarding the C:N ratio of Cretaceous vegetation and soils.

Through the use of the coupled LMDZ and ORCHIDEE models, our approach includes the pivotal coupling between atmosphere and vegetation. However by using fixed sea-surface temperatures, we neglect the feedbacks from the ocean-atmosphere coupling that could occur as a response to simulated changes in vegetation cover. Although sensitivity experiments with strong changes in vegetation suggested ocean feedbacks could play a significant role on the continental hydrological cycle (Davin and de Noblet-Ducoudré, 2010), our choice was motivated by (i) the will to focus on first-order continental processes, (ii) the computing cost required to equilibrate fully coupled simulations that typically require more than 3000 simulated years (Sepulchre et al., 2020) and the fact that comparable studies either used fixed-SSTs (Boyce and Lee, 2010; Lee and Boyce, 2010) or slab oceans (White et al., 2020).

Our parameterization of stomatal conductance in ORCHIDEE from Yin and Struik (2009) (Eq. (1)) is semi-empirical. A refinement of our modeling approach would be to use a stomatal conductance model based on optimisation theory, that explicitly describes the stomata functioning so as to optimise carbon gain against water loss (Medlyn et al., 2011; Buckley and Mott, 2013; Buckley, 2017). In particular, the model we use here simply links external forcing to leaf stomatal conductance by an empirical term of coupling fcpl that describe all the processes related to stomatal conductance. The $\alpha$ factor applied in Eq. (1) to the coupling factor (fcpl) does not fully represent the change in the maximal anatomic stomatal conductance as it should be considering changes inferred from vein density. We have seen that reducing the leaf hydraulic or photosynthetic capacity (or a combination of both) by a factor of 5 leads to a 3-fold decrease in leaf stomatal conductance. It emphasizes the need, in the future, to improve the parameterization of the stomatal conductance in global models by explicitly modeling both structural and dynamical conductance. Such a parameterization already exists for individuals plants (Dow et al., 2014), growing under controlled conditions, but not for land-surface models. Moreover, moving beyond PFT-based approaches towards truly trait-based ones would allow a more direct accounting for changes in functional traits.

Finally, we only consider the Cretaceous evolution of leaf hydraulic and photosynthetic capacity in this paper, as leaves are the end limit of plant water transport (Brodribb et al., 2007; Brodribb and Feild, 2010; Feild et al., 2011a, b). However, root thickness and wood structure for instance have evolved towards greater hydraulic efficiency (Brundrett, 2002; Wheeler and Baas, 2019). A comprehensive approach of the evolution of the plant hydraulic system would require to consider the entire soil-plant-atmosphere continuum, based on coupled resistances. Future studies would benefit from implementing a full hydraulic architecture of plants in the land surface model.

## 5 Conclusions

In line with recent studies focusing on Paleozoic vegetation transitions (White et al., 2020; Richey et al., 2021), the purpose of our study is to better represent past vegetation in earth system models by emulating "paleo-traits" in the vegetation parameterizations. Our approach involves an atmosphere-vegetation model, which couples stomatal conductance and carbon assimilation, motivated by an ecophysiological model based on angiosperm fossil records. Here, it allows us to evaluate three different pale-

ovegetation prescriptions under two end-member scenarios of $pCO_2$ for the Cretaceous. We show that the simulated vegetation cover, transpiration rate and water use efficiency are sensitive to the paleovegetation trait prescribed. Only accounting for leaf hydraulic capacity reduction provides no significant change in LAI, GPP and transpiration, while slightly increasing WUE at high $pCO_2$. In contrast, global transpiration decreases at low $pCO_2$ because of the positive feedback between LAI and stomatal conductance. On the other hand, only accounting for leaf photosynthetic capacity reduction gives a substantial decrease or even a collapse of vegetation at high or low $pCO_2$ respectively, which is in contradiction to the fossil record. Combining a reduction of leaf hydraulic capacity with that of photosynthetic capacity does not affect the plant productivity and LAI at high $pCO_2$ while vegetation collapses at low $pCO_2$. All the results in combination demonstrate that under high $pCO_2$ the reduced stomatal conductance of the proto-angiosperm vegetation is not a limiting factor on productivity. It also shows that high values of $V_{cmax}$ as observed in modern angiosperms do not enhance plant productivity, whereas maintaining the high $V_{cmax}$ likely requires higher leaf nitrogen concentration and higher energy demand. Therefore, a combination of lower-than-modern leaf hydraulic and photosynthetic capacities seems the most realistic physiological parameterization for proto-angiosperms in the specific high $pCO_2$ context of the Cretaceous. This is supported by evidence of coevolution inferred from previous studies (Franks and Beerling, 2009a, b) and the ratio of stomatal conductance between modern angiosperms and gymnosperms from in-situ experiments (Lin et al., 2015). Our results are also consistent with recent studies on coordination theory (Maire et al., 2012; Stocker et al., 2020).

Our study also suggests that proto-angiosperm vegetation with low leaf hydraulic and photosynthetic capacities was adapted to high $pCO_2$, where the combination of both physiological constraints nonetheless allowed high productivity and WUE. Conversely, it would not have been able to exist under low $pCO_2$, where we simulated a collapse of GPP under such physiological parameter configuration. Modeling the full coupling between GPP and stomatal conductance allows to understand why increasing both structural conductance and maximum photosynthetic capacity, even at an expense of a possible increasing water loss, was a selective advantage with decreasing $pCO_2$ and is the likely explanation for observed increasing structural conductance of angiosperms since the Cretaceous, consistent with previous studies (Franks and Beerling, 2009a, b; Boyce and Zwieniecki, 2012; Brodribb and Feild, 2010). From a low stomatal conductance (low vein density and $V_{cmax}$) similar to that of gymnosperms under high $pCO_2$, angiosperms evolve towards a high stomatal conductance (high vein density and $V_{cmax}$) to counteract the effect of the $pCO_2$ decrease on carbon assimilation.

While this study provides clues on how to account for angiosperm evolutionary traits in paleoclimate simulations, further work is needed to assess the potential climate effects of the Cretaceous angiosperm leaf evolution, especially on the hydrological cycle and the energy balance at the land surface. Furthermore, allowing dynamic vegetation would be an important future refinement of this research to model feedbacks between vegetation and climate. Also, replacing PFT morpho-physiological traits by species-specific traits (Kattge et al., 2020) as it has already been done for the aDGVM2 model (Scheiter et al., 2013), the LPJmL-FIT model (Sakschewski et al., 2015) or the JEDI model (Pavlick et al., 2013), allows plant communities to be assembled based on how plants with different trait combinations perform under a given set of environmental conditions. This way, changes in functional traits would be more directly taken into account. With such a dynamical vegetation model, we may question the veracity of the reciprocity with the human-induced increase in $pCO_2$. Building leaves with high vein density and

$V_{cmax}$ induces an investment in nutrients and energy (Chapin et al., 2011; Fiorin et al., 2016; Beerling and Franks, 2010) that could be an extra cost for plants without benefit under a higher $pCO_2$. That would suggest that plants should evolve back to a reduced stomatal conductance and $V_{cmax}$. However, the pace of the current increase in $pCO_2$ is dramatically higher than the million-year time scale of Cretaceous changes, and likely incompatible with a genetic plant adaptation. Whether extant plants will be able to adjust their physiological and morphological traits or not to the human perturbation is a challenging question for future studies.

*Code availability.* The code is available as part of the IPSL-CM5A2 earth system model, that has been made available by Sepulchre et al. (2020). The code can be retrieved through svn, with the following command lines:

svn co http://forge.ipsl.jussieu.fr/igcmg/svn/modipsl/branches/publications/IPSLCM5A2.1_11192019/

See also: https://gmd.copernicus.org/articles/13/3011/2020/#section11

The ORCHIDEE code, that has been modified for this study, is available as well through svn:

svn co https://forge.ipsl.jussieu.fr/orchidee/browser/branches/publications/ORCHIDEE_IPSLCM5A2.1.r5307. The login/password combination requested at first use to download the ORCHIDEE component is anonymous/anonymous.

*Data availability.* We provide the simulation outputs in the online repository : http://doi.org/10.5281/zenodo.5517167

*Author contributions.* JB, PS, NVIO and NVUI designed the work. Under supervision of PS, NVIO and NVUI, JB performed the simulations, processed the data, wrote the manuscript and drew the figures, except Fig. 1 drawn by PS. All authors reviewed and approved the manuscript.

*Competing interests.* The authors declare that they have no conflict of interest.

*Acknowledgements.* We thank the two anonymous reviewers for their meticulous work that helped to improve this manuscript a lot. This work was granted access to the HPC resources of TGCC under allocation 2019-A0090102212 made by GENCI. JB, NVIO and NVUI are supported by the *Commisariat à l'énergie atomique et aux énergies alternatives* (CEA). PS is supported by *Centre national de la recherche scientifique* (CNRS).

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
