# Peer review of "The Cretaceous physiological adaptation of angiosperms to a declining pCO2 : a trait-oriented modelling approach"

_Biogeosciences, 2021_

## Author Comment (AC1)

**Responses to RC1**

Dear referee,

Thank you very much for your reading. We really appreciate your contribution to helping us to improve our manuscript with helpful suggestions and comments. Here below, your comments are highlighted in black and italicised and our replies are highlighted in blue. We sincerely hope that you are satisfied with our replies and our proposed changes.

Sincerely yours,

Julia Bres and co-authors.

**General Comments:**

*I found this paper to be very interesting and a good first step in estimating the impact of the evolution of angiosperms. Though there are limitations to this study, I think that they were very nicely and plainly listed and acknowledged as such. This study will an important contribution to the continuing attempts to better represent paleo vegetation in climate modeling.*

*I think the paper could use some work with the grammar. There are a lot of sentences that are missing needed commas or need be rearranged for clarity. I have attempted to point these out when possible.*

*I think this paper fits well within the scope of Biogeosciences and would recommend its publication pending minor to moderate revisions.*

Thank you for these comments. In the following, we provide detailed replies and corrections for RC1 concerns. We also rephrased several sentences following the referee suggestions to improve the grammar.

**Specific Comments:**

*Abstract:*

*Line 14 and 15: "Stage"? Maybe "state" would be better.*

We modified the text accordingly.

*Line 15: Irrigated? That sounds off. Maybe say that the leaves are kept more flush with water?*

We rephrased to "more densely veinated".

*Introduction:*

*Line 54: Not sure if this falls into "specific" or "technical", but surely there is a better phrase than "a lot"? Maybe "Even if other functional traits evolved during the angiosperm radiation, ..." is better ?*

We agree and we rephrased accordingly.

*Methods:*

*Lines 98-104: This sentence is very hard to comprehend. Way too much information in one sentence. I recommend splitting it up. Also, should "a day respiration" be "daily respiration"?*

We agree. We rephrased accordingly. Now, you can read : "Within the model used in ORCHIDEE (Yin and Struik 2009), leaf operational stomatal conductance to $H_2O$ ($g_s$ in mol m$^{-2}$[leaf] s$^{-1}$) depends on the net carbon assimilation (A in molCO$_2$ m$^{-2}$[leaf] s$^{-1}$), the daily respiration ($R_d$ in molCO$_2$ m$^{-2}$[leaf] s$^{-1}$), the intercellular $CO_2$ partial pressure ($C_i$ in bar), the $C_i$-based $CO_2$ compensation point in the absence of $R_d$ ($C_i^*$ in bar) and the residual stomatal conductance to $H_2O$ ($g_0$ in mol m$^{-2}$[leaf] s$^{-1}$), to account for a non-zero conductance when the carbon assimilation is zero (Farquhar, Caemmerer, and Berry 1980; Ball, Woodrow, and Berry 1987; Yin and Struik 2009). Finally, the leaf operational stomatal conductance to $H_2O$ is modulated by a factor fcpl, describing the strength of the coupling between A and $g_s$, which is function of the leaf-to-air vapor pressure deficit (kPa) and that we will further name "leaf hydraulic capacity" ..."

*Lines 106-108: Again, this sentence is hard to comprehend. Suggest changing "leaves morphologic and physiologic traits" to "the morphologic and physiologic traits of leaves" for clarity. Otherwise, this sentence should be split for clarity.*

The sentence is now rephrased as : "The semi-empirical formalism of $g_s$ allows to account for both the structural conductance, linked to the morphologic and physiologic traits of leaves (e.g. $D_v$, $D_s$ and S) developed on the long-term plant evolution history, and the dynamical conductance, related to the short-term stomatal opening and closing depending on the environment."

*Lines 118-132: This section is much more clear and should be a model for sentences listed above.*

Thank you for this remark, we have now used this section as a model.

*Line 171: This sentence is awkward and should be changed to "A lower value for y indicates a lower stomata-to-vein distance and a higher...." Or something similar.*

It now reads "A lower value for y indicates a lower stomata-to-vein distance and a higher $g_{anat}^{max}$."

*Line 187: Why was solar radiation set at 99%? For instance, in the Late Paleozoic, solar radiation is thought to be 3% lower based on the modeling of Crowley and Baum 1992. Is there similar modeling evidence for the Cretaceous. If so, it should be cited. If not, it should be justified.*

From Gough formulation (1981), the solar constant rose from 98.76 to 99.4 % of its present-day value between 144 Ma and 65 Ma (the lower and upper bounds of the Cretaceous). We actually underestimated it by setting it at 1345 W.m-2, which is 98.53 % of its present-day value. Based on the values depicted in Laugié et al. (2020) with the IPSL model, we estimate that this 0.5 % underestimation should translate to a homogeneous cooling of no more than 0.8°C and should not alter our results.

*Line 201: I'm not sure I agree with the 5-fold lower photosynthetic and hydraulic capabilities of pre-angiosperms because it is based on the notion that vein density trumps all other physiological attributes, but for the purpose of a sensitivity analysis and your paper, it is appropriate and important. Carry on.*

Yes, it is a strong assumption that we make. We clarify that point in the introduction: "Paleobotanical data show that several traits from the soil-plant-atmosphere continuum have evolved during the Cretaceous. Specifically, root thickness and wood structure have evolved towards greater hydraulic efficiency (Brundrett 2002; Wheeler and Baas 2019]. In this paper, we focus on the upper part of this continuum, considering changes in leaf stomatal conductance and leaf photosynthetic capacity."

*Results:*

*Line 219: "3-fold" instead of "3-time".*

Corrected.

*Line 223-224: The fact that gymnosperm function was not accessed is a problem for me. Yes, without a doubt, angiosperms have displaced gymnosperms as the dominant plant through time, but they still, to this day, occupy vast and important parts of the Earth's surface. Understanding their response in terms of a global climate model is important for the whole story of the increase in angiosperm dominance across the earth surface. Ultimately, judging the effect of angiosperms in the face of no change in gymnosperms seems to be missing a large part of the story. However, given your stated purpose, and the fact that needleleaf gymnosperms are not widespread in your paleovegatation distribution map, it is appropriate.*

Indeed, explaining the gymnosperm response to $pCO_2$ variations, as well as explaining the long-term persistence of gymnosperms, is out of the scope of our study. Here, not addressing the case of the long-term physiological evolution of gymnosperms is justified by the fossil data that shows their vein density remains constant over the Cretaceous (Feild et al. 2011). To make it clearer for the reader, we have added the mean value (one standard deviation) for gymnosperm vein density and maximal anatomic stomatal conductance to $H_2O$ on figure 2.

[Figure]

(a) Time evolution of fossil vein densities (mm mm$^{-2}$, y-axis) adapted from Feild et al. 2011 and the corresponding maximal stomatal conductance to $H_2O$ (mol m$^{-2}$ s$^{-1}$, filled circles) calculated by the anatomic relationship developed by Brodribb et al., 2007 and Brodribb et al., (2010). (b) Subdivision of maximal stomatal conductance to $H_2O$ (mol m$^{-2}$ s$^{-1}$) by time periods. The solid line is the median, the dotted lines are the first and third quartiles, and the point is the mean value for each time period.

*Line 255-257: Doesn't this show the importance of highlighting the modeled change in angiosperms vs. gymnosperms? Gymnosperms are showing the resilience of coming to dominance during the low $CO_2$ of the Late Paleozoic Ice Age.*

We agree. Although the resilience of gymnosperms is out of the scope of our study, future studies could focus on traits linked to fire resistance, as they have been shown to be crucial to explain the long-term persistence of some gymnosperm clades - namely the Pinus clade (Singh et al. 2018).

*Line 263-264: Soil water is always the limiting factor, especially in the tropics.*

We show more in the text that this is what we were expecting.

*Line 271-276: Indeed. Leaves do not function in a vacuum. Without accessing the whole canopy, we cannot truly trust the implications.*

Indeed, it is one of the strengths of our modelling approach.

*Discussion:*

*Line 478: I do not understand this sentence. Are you saying vein density is stomatal size and density? Or are you saying that vein density and stomatal size and density need to be accounted for? Clarify*

Here, we mean that the $\alpha$ factor applied in Eq. (1) to the coupling factor (fcpl) does not fully represent the change in the maximal anatomic stomatal conductance inferred from the morphological leaf traits, such as $D_v$ (but another approach could consider $D_s$ and S). Indeed, we have seen that $g_s$ is 3-fold lower for

the perturbed experiments compared to the reference ones, while we explicitly account for a 5-fold decrease in fcpl and $V_{cmax}$, to mimic maximal anatomic stomatal conductance variation. One way to improve our parameterization of proto-angiosperm vegetation in land surface models is to explicitly model both structural and dynamic conductance, and apply the 5-fold decrease to the structural conductance, as it represents leaf traits evolution. We acknowledge that the sentence was not clear and we rephrased it : " The $\alpha$ factor applied in Eq. (1) to the coupling factor (fcpl) does not fully represent the change in the maximal anatomic stomatal conductance as it should be considering changes inferred from vein density. We have seen that reducing the hydraulic or photosynthetic capacity by a factor of 5 (or a combination of both), leads to a 3-fold decrease in leaf stomatal conductance. It emphasizes the need, in the future, to improve the parameterization of stomatal conductance in global models by explicitly modelling both structural and dynamic conductance."

***Technical corrections:***

*Introduction:*

*Line 34: Should there be a space between "95" and "%"? I was under the impression that there should not.*

We will check with the copy-setting board. According to Biogeosciences guidelines, "Spaces must be included between number and unit (e.g. 1 %, 1 m)" for figure content guidelines but nothing is mentioned for manuscript composition.

*Line 35: Change "at the expanse" to "at the expense".*

Corrected.

*Line 42: This sentence would be clearer with "i.e., stomata" was parenthetical.*

We agree, we have added the parentheses.

*Line 61: "of $H_2O$", not "to $H_2O$".*

Stomatal conductance to $H_2O$ or to $CO_2$ is widely used, for instance in Franks et Beerling (2009), White et al. (2020) or Lammertsma et al. (2011). We would prefer keep "to $H_2O$".

*Line 63: seems as though "simulation design" would be better than "choices made"?*

We agree, it is now rephrased.

*Lines 98-104: In line with my earlier comment on this sentence: To improve this sentence, or sentences after editing, I suggest placing the abbreviation for parameters inside the parentheses with the units.*

Indeed, it is clearer with parentheses.

*Line 182: Need comma after "both".*

Corrected.

*Figure 3 caption: Should this read "fixed through time"?*

Yes, it is now changed.

**References**

Ball, J Timothy, Ian E Woodrow, and Joseph A Berry (1987). "A model predicting stomatal conductance and its contribution to the control of photosynthesis under different environmental conditions". In: *Progress in photosynthesis research*. Springer, pp. 221–224. DOI: `https://doi.org/10.1007/978-94-017-0519-6_48`.

Brodribb, Tim J and Taylor S Feild (2010). "Leaf hydraulic evolution led a surge in leaf photosynthetic capacity during early angiosperm diversification". In: *Ecol. Lett.* 13.2, pp. 175–183. DOI: `https://doi.org/10.1111/j.1461-0248.2009.01410.x`.

Brodribb, Tim J, Taylor S Feild, and Gregory J Jordan (2007). "Leaf maximum photosynthetic rate and venation are linked by hydraulics". In: *Plant Physiol.* 144.4, pp. 1890–1898. DOI: `https://doi.org/10.1104/pp.107.101352`.

Brundrett, Mark C (2002). "Coevolution of roots and mycorrhizas of land plants". In: *New Phytol.* 154.2, pp. 275–304. DOI: `https://doi.org/10.1046/j.1469-8137.2002.00397.x`.

Farquhar, Graham D, S von von Caemmerer, and Joseph A Berry (1980). "A biochemical model of photosynthetic CO 2 assimilation in leaves of C 3 species". In: *Planta* 149.1, pp. 78–90. DOI: `https://doi.org/10.1007/BF00386231`.

Feild, Taylor S et al. (2011). "Fossil evidence for Cretaceous escalation in angiosperm leaf vein evolution". In: *P. Natl. Acad. Sci. USA* 108.20, pp. 8363–8366. DOI: `https://doi.org/10.1073/pnas.1014456108`.

Franks, PJ and DJ Beerling (2009). "CO2-forced evolution of plant gas exchange capacity and water-use efficiency over the Phanerozoic". In: *Geobiology* 7.2, pp. 227–236. DOI: `https://doi.org/10.1111/j.1472-4669.2009.00193.x`.

Gough, DO (1981). "Solar interior structure and luminosity variations". In: *Physics of solar variations*. Springer, pp. 21–34. DOI: `https://doi.org/10.1007/978-94-010-9633-1_4`.

Lammertsma, Emmy I et al. (2011). "Global CO2 rise leads to reduced maximum stomatal conductance in Florida vegetation". In: *Proceedings of the National Academy of Sciences* 108.10, pp. 4035–4040. DOI: `https://doi.org/10.1073/pnas.1100371108`.

Laugié, Marie et al. (2020). "Stripping back the modern to reveal the Cenomanian–Turonian climate and temperature gradient underneath". In: *Climate of the Past* 16.3, pp. 953–971. DOI: `https://doi.org/10.5194/cp-16-953-2020`.

Singh, Surendra P et al. (2018). "Insights on the persistence of pines (Pinus species) in the Late Cretaceous and their increasing dominance in the Anthropocene". In: *Ecology and evolution* 8.20, https://doi.org/10345–10359. DOI: `https://doi-org.insu.bib.cnrs.fr/10.1002/ece3.4499`.

Wheeler, Elisabeth A and Pieter Baas (2019). "Wood evolution: Baileyan trends and functional traits in the fossil record". In: *IAWA journal* 40.3, pp. 488–529. DOI: `https://doi.org/10.1163/22941932-40190230`.

White, Joseph D et al. (2020). "A process-based ecosystem model (Paleo-BGC) to simulate the dynamic response of Late Carboniferous plants to elevated O2 and aridification". In: *Am. J. Sci.* 320.7, pp. 547–598. DOI: `https://doi.org/10.2475/09.2020.01`.

Yin, X and PC Struik (2009). "C3 and C4 photosynthesis models: An overview from the perspective of crop modelling". In: *NJAS-Wagen. J. Life Sc.* 57.1, pp. 27–38. DOI: `https://doi.org/10.1016/j.njas.2009.07.001`.

---

## Author Comment (AC2)

**Responses to RC2**

Dear referee,

Thank you very much for your careful evaluation of this manuscript and the constructive critique which will guide our revisions and greatly improve this paper. Here below, your comments are highlighted in black and italicised and our replies are highlighted in blue. We sincerely hope that you are satisfied with our replies and our proposed changes.

Sincerely yours,

Julia Bres and co-authors.

*The authors present a three-parameter sensitivity study involving atmospheric $CO_2$ concentration, maximum upper stomatal conductance constraint (physiological constraint on dynamic stomatal conductance), and photosynthetic capacity via adjustment of $V_{cmax}$. They motivate this study using the example of angiosperm evolution and associated adjustment of physiological traits during the Aptian age of the Lower Cretaceous. The study is interesting and worthwhile publishing, as it not only contributes to an improved understanding of the physiological adaptations and adjustments that accompanied the angiosperm evolution and radiation, as well as possible driving factors behind these physiological adjustments, but also offers insights on model sensitivity with regard to these parameters. The evolution of higher vein densities and associated increase in structural stomatal conductance concomitant with the angiosperm radiation must have been functionally beneficial and is therefore worthwhile to be studied in more detail in order to improve understanding of associated processes and effects.*

*The authors use the ORCHIDEE DGVM/land surface model and LMDz atmosphere general circulation model versions embedded in the IPSL-CM5A2 model, which implies that the underlying vegetation model is not suitable to directly simulate functional traits and trait diversity. The authors circumvent this constraint by mimicking the traits using parameters available in their model. This approach is valid, and part of the limitations as well as suggested improvements are discussed in the discussion section. However, it should be made more clear also with respect to the title of the study that it is not a truly trait-based modeling study.*

We thank the reviewer for these comments. We modified the title (see the specific section) to account for these comments.

*With regard to stomatal regulation on daily basis (operational/dynamic stomatal regulation), I would like to*

*have some clarification on the influence of water stress, because it did not become clear to me whether and how water constraint will down-regulate stomatal conductance.*

First of all, stomatal regulation is calculated on a half hourly basis. Then, water stress on stomatal conductance is indirectly accounted for through a limitation on $V_{cmax}$. The latter is supported by Keenan et al., (2009 & 2010) and Egea et al. (2011). The stress factor is an empirical factor between 0-1 calculated from soil humidity. This point has been clarified in the text.

*The manuscript is structured clearly in most parts, the language is appropriate and understandable (although the English would benefit from additional grammar and style corrections). However, I have a variety of comments and suggestions on how the manuscript could be improved further, which I am listing in the following.*

**Specific comments for the authors:**

**Title**: *I'm not sure whether the title is appropriate. Personally, I find it a bit misleading, because it made me expect a study using a trait-based DGVM that explicitly considers plant functional traits. Only after reading the manuscript it became clear that it is actually a three-parameter sensitivity study that indirectly emulates functional traits and is therefore limited compared to the full range of possibilities offered by trait-based adaptive DGVMs. I therefore suggest adjusting the title to make this more clear.*

Thank you for this comment, the reader's expectations should be more in line with the content of the article. We suggest replacing the original title by: "The Cretaceous physiological adaptation of angiosperms to a declining $pCO_2$: a modeling approach emulating paleo-traits".

**Introduction**:

*l. 41 "showing a strong correlation with vein density": clarify in which direction: positive correlation (i.e., higher vein density correlated to larger and more dense stomata), or negative (e.g., more but smaller stomata associated with higher vein density).*

Fossils show that high vein density is found for leaves with high stomatal density but with small stomata. We rephrased the sentence as: "Contemporaneous changes in stomatal density (size), hereafter called $D_s$ (S), are observed in the fossil record (De Boer et al. 2012) showing a strong positive (negative) correlation with vein density."

*l. 42 "veins allow the plants to efficiently transport water from the soil close to the site of transpiration": That's not only leaf veins, this is also dependent on xylem (type, diameter, reinforcement) in plant stems. The soil-plant-atmosphere continuum (SPA) also depends on the hydraulic capacity of the xylem and the associated suction pressure it can withstand before cavitation becomes substantial at high suction pressure (p50 concept). Xylem structure/anatomy also differs quite substantially between gymnosperms (pitted tracheids) and angiosperms (trachea/vessel elements). This is another trait that has implications for functionality with regard to water transport through the plant. It would be nice if you could at least write a little bit about that here, and whether and how this is accounted for in the ORCHIDEE model.*

We agree that the hydraulic architecture of the plant is also a main driver of the transpiration and that it also probably changed with time. Unfortunately this is not implemented in the standard version of ORCHIDEE used in the coupled model. Like in most of the DGVMs, this is only (very) indirectly taken into account through a hydric stress factor based on soil water content that controls stomatal closure. Then, in this paper we focus only on the foliar properties. But as suggested we added a discussion about this important point into the manuscript. We add a reference to SPA continuum and we discuss this limitation. We wrote: "Finally, we only consider the Cretaceous evolution of leaf hydraulic and photosynthetic capacity in this paper, as leaves are the end limit of plant water transport (Brodribb et al., 2007; Brodribb et al., 2010; Feild et al., 2011ab). However root thickness and wood structure have evolved towards greater hydraulic efficiency (Brundrett 2002; Wheeler and Baas 2019). A comprehensive approach of the evolution of the plant hydraulic system would require to consider the entire soil-plant-atmosphere continuum, and future studies would benefit from implementing a hydraulic architecture of plants in the land surface model."

*l. 43/44 "Dv is a reliable marker of hydraulic capacity": is Dv correlated with p50? I'd expect it to, at least to some degree.*

We were here talking about hydraulic capacity into the leaf as Dv here represents the vein density into the leaf. But we agree that leaf being the end part of the full chain of conductivity of water from soil to the atmosphere into the plant, change in Dv should participate to the change in the response curve of percent loss of conductivity to xylem pressure and then to the xylem pressure associated to p50.

*l. 51 "bolster the current hydrological cycle": Find a more appropriate verb? Not sure what you mean to say exactly.*

We replaced "bolster" with "strengthen".

*l. 54/55 At least mention the other traits associated with the SPA continuum that you are omitting, because it is the combination of all traits in the SPA continuum that mitigate the water flux between soil, plants and atmosphere.*

We agree. We enhanced this section adding references to paleo-traits in the SPA continuum that are known to have evolved during the Cretaceous. It now reads: "Paleobotanical data show that several traits from the Soil-Plant-Atmosphere continuum have evolved during the Cretaceous. Specifically, root thickness and wood structure have evolved towards greater hydraulic efficiency (Brundrett 2002; Wheeler and Baas 2019). In this paper we focus on the upper part of this continuum, considering changes in leaf stomatal conductance and photosynthetic capacity."

*l. 63 "Depending on the choices made": which choices?*

Here choices refer to the three parameterizations (NOANGIOh, NOANGIOp, and NOANGIOhp) depending on values affected to $\alpha$ and $\beta$, choices explained in the Sect. "Experimental setup". We rephrased the sentence that now reads: "[...] they can be parameterized to represent the changes in stomatal conductance

that occured with the angiosperm radiation. Depending on their design, these parameterizations can reflect changes on plant traits controlling hydraulic capacity, photosynthetic capacity or both."

*l. 67/68 "Moreover, atmospheric $pCO_2$ is known to control the degree of stomatal opening and closing at very short-term": Are you talking about structural stomatal opening here, or about operational opening? Because operational opening would not be controlled by $pCO_2$ alone, but jointly with water availability in relation to atmospheric demand (vapor pressure deficit, VPD). This co-regulation by C-demand for assimilation vs. water demand by the atmosphere indeed happens on very short time scales, whereas structural stomatal changes occur on evolutionary time scales.*

Thanks for the remark. We are talking about operational opening and obviously $pCO_2$ is only one factor that drives the operational conductance, all the others factors are also taken into account in the model. But indeed this was not clear in the text so we replace it by: "Moreover, atmospheric $pCO_2$ is known to be one of the drivers that control the stomatal opening and closing at very short-term".

*l. 75-77: Just a comment:This is an interesting question - why did the higher Dv evolve and become abundant? It must have been beneficial, otherwise it would not have spread and persisted. It's therefore worthwhile to investigate the associated effects and dedicate a study to that topic. In particular if this helps to better understand connections between physiology and function that may not be as well understood as required to adequately represent plant functional traits in vegetation models. It is important to translate observed plant traits, whether they are paleo-traits or contemporary, into functional meaning in order to understand and quantify their implications for vegetation reactions on environmental conditions.*

Indeed, this is one of the important conclusions of the paper just using a vegetation model: mimicking the change in higher Dv by changing the stomatal conductance shows that lower conductivity of first angiosperms does not affect their productivity under high $CO_2$. In contrast, it drastically reduces productivity under current $CO_2$ conditions. Hence, increasing Dv and the number of stomata was a selective advantage since decreasing photosynthesis efficiency by reduced $CO_2$ needs a higher surface of exchange with atmosphere even if it implies an increasing transpiration cost.

*What I am missing at the end of the introduction is a statement listing your research questions or research hypotheses. I suggest to add such an explicit statement of your main research questions or hypotheses here, because it makes it easier to evaluate your results in relation to the research goals of the study.*

We agree and we have rephrased several sentences at the end of the introduction in order to highlight our goals and hypotheses.

**Methods:**

*l. 87/88 Are the 4-meter soil depth a global parameter? Soil depth has a major influence on plant water availability as it defines the size of the bucket, especially in simple bucket model representations e.g., see Langan et al., 2017, Journal of Biogeography, doi:10.1111/jbi.13018). I suppose there is no possibility to infer soil parameters or soil depth from the Aptian time, so you have to make assumptions. Please also list*

*this as a limitation in the discussion where you talk about other limitations and uncertainties associated to your study. Also not mentioned: what kind of other soil parameters does ORCHIDEE require (soil texture, wilting point, ksat, etc), and how did you parameterize these parameters?*

Yes, the 4-meter soil depth is set globally. It is a refinement of the initial model which accounts for a 2-meter soil depth at the origin of dry biaises (see the discussion about this choice in Dufresne et al., (2013)). More information about how the bucket model works can be found in Sepulchre et al. (2020): "Water is redistributed between the two layers through a downward flux parameterized following the early ideas of Choisnel (Choisnel et al., 1995; Ducharne et al., 1998). Rain falling from the canopy feeds the upper layer that loses water both by root extraction and soil evaporation, whereas water storage in the bottom layer decreases only as a function of root extraction (Guimberteau et al. 2014). When total soil moisture storage reaches the maximum water storage, the excess water amount is converted to runoff". The influence of soil depth (especially the change from 2m to 4m) is partly limited by the fact that plant available water is convoluted by root profile and then mainly limited to tropical forest which have very deep roots. This version is adequate to simulate evapotranspiration fluxes consistent with the data (Guimberteau et al. 2014). Other soil parameters include soil texture (silt/sand/clay fraction) and soil color, which determines the visible and infra-red albedo of the bare soil. Although paleosoil texture data do exist for the Cretaceous, they are regional, and reconstructing Cretaceous soil texture at the global level would have required too much effort for an uncertain result, since sensitivity studies carried on present-day have shown a rather weak sensitivity of the terrestrial water budget to soil texture in the ORCHIDEE model (Tafasca et al., 2020). Hence, we globally set this parameter to an average value corresponding to loam in Zobler classification (0.39 % silt, 0.43 % sand and 0.18 % clay). Soil color determines the bare soil albedo and is set globally at 0.16 (Wilson and Henderson-Sellers 1985). We added a paragraph on that point: "Although scattered paleosoil texture data do exist for the Cretaceous, sensitivity studies carried on present-day have shown a rather weak sensitivity of the terrestrial water budget to soil texture in the ORCHIDEE model (Tafasca et al., 2020). Thus, we set this parameter worldwide to an average value corresponding to loam in Zobler classification (Zobler 1986): 0.39 % silt, 0.43 % sand and 0.18 % clay. Soil color determines the bare soil albedo and is set globally at 0.16 (Wilson and Henderson-Sellers 1985)."

*l. 92: "while carbon-related slow processes are computed on a daily basis": What carbon-related processes are these in detail that you define as "slow processes"? The temporal resolution for handling of C-related processes varies between DGVMs. Some DGVMs do carbon allocation and respiration on daily basis, others on even coarser time scales (monthly, annual).*

The highest timestep in ORCHIDEE is 30 minutes. At this time step we calculate all the biophysical processes but also photosynthesis and respiration. Then "slow processes" in the model are processes such as carbon allocation and leaf phenology, which are calculated on a daily basis. We have added parentheses with such processes to be clearer.

*l. 98-104: "leaf operational stomatal conductance to $H_2O$, $g_s$ (mol $m^{-2}$ [leaf] $s^{-1}$), depends on the net carbon assimilation A...": Does the model also account for water limitation effects on stomatal regulation? When water demand required by the atmosphere, via water vapor deficit (VPD) cannot be met by the quantity of water that can be supplied via the SPA continuum, stomata should close in response to the water shortage.*

*In more detail, this means that as long as the water demand required by the atmosphere can be met, the stomata regulation can be driven by the carbon side, i.e., the carbon demand of photosynthesis, as represented by Ci, under the given limiting conditions to photosynthesis (Jc, Je, Js). But if the water demand by the atmosphere cannot be served any more by the soil-plant continuum due to low water content/resistance that exceeds the transport capacity required to fulfill the atmospheric demand, this should also trigger stomatal closure, at least up to the point where stomatal conductance just about equals the maximum water loss possible under the water constraint, i.e., the quantity of water that can be provided through the SP-system. Under such conditions, photosynthesis, via water-limitation-induced stomatal closure or partial closure, should be constrained by stomatal conductance, rather than stomatal conductance being constrained by assimilation capacity. This water-stress-induced stomatal closure will not only down-regulate photosynthesis under water stress, but at the same time reduce transpirational cooling of leaves as latent heat flux decreases and sensible heat flux concomittantly increases, which leads to an increase in leaf temperature under water stress and impacts temperature-dependent $V_{cmax}$. What is required to fully account for this is a photosynthesis routine that simultaneously iteratively solves for stomatal conductance, assimilation, transpiration and leaf temperature under the constraints imposed by energy balance and system-internal resistances. In this context, also see the publications by Schymanski & Or (2017, Hydrology and Earth System Sciences, doi:10.5194/hess-21-685-2017) and Tuzet et al. (2003, Plant, Cell and Environment, doi:10.1046/j.1365-3040.2003.01035.x).*

As specified previously, water stress acts directly on $V_{cmax}$, as supported by previous studies Keenan et al., 2009 & 2010 ; Egea et al., 2011). We acknowledge that in the standard version of ORCHIDEE used here, like for most DGVMs, we do not represent the full soil/plant/atmosphere continuum, especially the water transportation in the plant. However the water limitation is taken into account through an empirical factor calculated from soil water that allows indirectly to limit stomatal conductance independently of the other factors (i.e photosynthesis, $CO_2$ and VPD). Likewise the temperature taken into account for photosynthesis is the surface temperature. Although it is a skin temperature and not the actual leaf temperature, the feedback of stomatal closure on photosynthesis through change in temperature is taken into account. The parameter fcpl that couples the operating stomatal conductance to external forcings takes into account the VPD cf Yin et Struik (2009). We add the formulation in the text.

*General note: Variable Symbols and abbreviations: it would be good to have a reference table for all used variable symbols and abbreviations, so that these are collected in one place to look them up as necessary.*

Thank you for this suggestion, we agree and the table for abbreviations is now included in the manuscript (Table 1).

*l. 110, Eqn. 2: How about substrate limitation (often abbreviated as Vs, or Js)?*

In the formulation of Yin et Struick (2009) used for photosynthesis, they assume no substrate limitation, only the most limiting factor between $V_c$ and $V_j$.

*l. 114: How about leaf nitrogen contents, or are these tied to leaf age, i.e., leaf age is a surrogate for N_leaf? (see Sakschewski et al., 2015, GCB, doi.org/10.1111/gcb.12870). What makes $V_{cmax}$ different among PFTs? Temperature-dependence relationship/optimum temperature?*

We simply mean that $V_{cmax}$ dynamically varies seasonally with the age of leaves (we consider 4 cohorts of leaves, see Krinner et al. 2005). The $V_{cmax}$ at 25°C is prescribed for each PFT based on compilation of data but the temperature-related photosynthesis parameters are set constant for all C3 species. $V_{cmax}$ and leaf lifespan are prescribed independently also based on literature and then leaf age is not a surrogate of N_leaf even if obviously, because of the leaf economic spectrum there is an inverse relationship between leaf lifespan and $V_{cmax}$. However, in the experiment when we modify the $V_{cmax}$, there is no change in leaf lifespan.

*l. 122/123 "This factor depends on both the soil moisture and the root profile given per PFT, trees having deeper roots than grasses.": Do you, in ORCHIDEE, account for plant-internal resistances to water transport (coupling of resistances associated with soil-root transition, root-internal transport resistances, stem-internal resistances, stomatal resistance, boundary layer resistances)?*

We do not. This is something under development but the standard version used in the coupled model takes into account only stomatal resistance, not resistances to water transport. We just use a stress factor related to soil water that takes into account an idealized root profile that depends on the PFT (to represent, for instance, the fact that hydric stress arises sooner in herbaceous vegetation than for trees).

*l. 135: I like the idea of separating into structural vs. dynamical/operational resistance due to stomatal conductance. Here, I would explicitly state that the structural resistance defines the upper boundary condition that constrains the dynamical resistance, i.e., stomatal regulation varies very quickly based on daily environmental conditions (C-demand by assimilation as a function of light availability and temperature, water-stress driven regulation of stomatal conductance), but the maximum aperture and associated conductance that can be assumed dynamically is defined by the structural constraint (if I understood that correctly).*

That is correct, and we've added a sentence accordingly.

*l. 137, nitrogen stress: state here that you used ORCHIDEE, not ORCHIDEE-CN for this study, i.e., that you did not account for nitrogen-cycle-related aspects. Otherwise, in O-CN, would leaf nitrogen content influence $V_{cmax}$ (see Sakschewski et al., 2015, https://doi.org/10.1111/gcb.12870)?*

It is now specified here that we do not use ORCHIDE-CN because of limitations (see limitation section of the manuscript). In O-CN or ORCHIDEE-CN, indeed, $V_{cmax}$ is a function of leaf N content ($V_{cmax}$=Leaf N content * NUE where NUE is the Nitrogen Use Efficiency ($\mu$mol $CO_2$ m$^{-2}$ s$^{-1}$ (gN)$^{-1}$).

*l. 151: "At the canopy level (Fig. 1), the canopy conductance $g_c$ depends on both leaf level conductance $g_s$, and LAI.": How about coupling to boundary layer conductance? That should have an influence as well?*

The atmospheric boundary layer conductance is taken into account via $r_a$ (Eq. (5)). The leaf boundary layer conductance is taken into account in $C_i$ formulation, that depends also on ambient air $CO_2$ partial pressure, stomatal conductance and carbon assimilation (see Yin et Struik, 2009, eq. 16). We have now specified that the boundary layer conductance is accounted for in the computation of the transpiration.

*l. 154 "A low soil water content will induce a water stress limiting the $V_{cmax}$ and $V_{jmax}$ that will indirectly also reduce $g_s$.": Is that how water stress is accounted for in the model? I don't think the mechanistic effect of soil water stress on stomatal closure is via $V_{cmax}/V_{jmax}$ – do you have a reference for that? As I said before, decreasing soil water contents should increase the resistance in the SPA system and limit the amount of water that can be transpired compared to the actual atmospheric demand, and the maximum aperture under water stress should be defined by the quantity that can be provided through the SPA-resistance, i.e., it should be maximum wide enough to allow no more transpiration than can be provided to be transpired.*

Actually several papers (for instance Zhou et al., 2014) show a limitation of hydric stress on both $V_{cmax}/V_{jmax}$ and stomatal conductance. In the original formulation of ORCHIDEE, the hydric stress was applied to $V_{cmax}/V_{jmax}$ only. But we acknowledge that it is not fully correct. In very recent versions of ORCHIDEE, we now apply it on both $V_{cmax}/V_{jmax}$ and $g_s$. But in the version used in the coupled model it is still the version with the impact on $V_{cmax}/V_{jmax}$ only. However, we made tests and because of the feedback between photosynthesis and stomatal conductance, there is very little difference in simulated photosynthesis and stomatal conductance using the two methods.

*l. 168: Why do you set VPD to a constant value? It ought to chance based on the relative humidity in the boundary layer of the leaf and the coupling between leaf boundary layer and canopy boundary layer (moisture transport away from the leaf)? I suppose the atmosphere model simulates changes in relative humidity/VPD?*

Indeed, the atmosphere model simulates change in VPD and the ORCHIDEE conductance model includes response to VPD as well (Eq. (2)). But in equation 8, from Brodribb and Feild (2010), we calculate the $g_{anat}^{max}$ that is a static long term value that cannot depend on a dynamical parameter. For this reason, VPD here is a mean standard value.

*l. 176-179: This should be made clear earlier on – that this is not a direct functional trait-based modeling study, but mimics functional constraints indirectly by constraining maximum $g_s$ with different caps depending on the structural conductance.*

We agree, and we have changed the title and several sentences in the manuscript to be clearer on that important point.

*l. 181/182: "...we performed climate model simulations...": What does not really become clear to me here is whether you modeled the vegetation dynamics of the land surface, or whether you described the PFT-distribution rather than letting it develop. I suppose the former, based on your statements in discussion and conclusion section. Please explicitly mention here, and also briefly justify why you refrain from modeling vegetation dynamics.*

Vegetation distribution is prescribed and is not dynamic (cf Fig. 3). That means that only the PFTs prescribed (see Table S2) can grow on a particular grid point. It is the first time that such sensitivity tests were made (i) with the ORCHiDEE model and (ii) for this deep time period so setting PFT distribution is a first step to understand plant physiology linked to angiosperm leaf evolution. This method avoids additional feedback linked to vegetation dynamics, that would make the mechanisms at work more complex to decipher. A

further study will use the DGVM to let vegetation freely grow, but we expect also technical difficulties linked to numerical instabilities in the atmosphere-vegetation coupling at such high $pCO_2$. It is a second reason why we hold with a fixed PFT distribution. We mention that: "We prescribe PFT distribution rather than activate the dynamical vegetation model to constrain our interpretations of the sensitivity experiments, avoid additional feedback linked to vegetation dynamics and potential numerical difficulties linked to atmosphere-vegetation coupling at high $pCO_2$."

*l. 185 (paleobotanical data, reference Sewall et al., 2007), and Figure 3: During the Aptian, there should be no C3 grasslands! Also, in the Sewall paper, I did not find anything about grass or grassland, or grassy biomes. C3 grass evolution likely did not really start before the end of the Upper Cretaceous, and their spread to dominance in open grassland biomes only happened during the Tertiary, to my knowledge. See, for example, the review paper by Caroline Stromberg (Evolution of Grasses and Grassland Ecosytems, 2011, Annu. Rev. Earth Planet. Sci., 2011 (39), 517-44), and chapter 5 in William Bond's recent book "Open Ecosystems – ecology and evolution beyond the forest edge (Oxford University Press)". In my opinion, having the C3 grass biome present and as widely spread at that time is a mistake, if the simulations are truly supposed to represent Aptian biomes.*

Thank you for this comment. We are aware and fully agree that grassland started to expand only during the Cenozoic. The challenge for our experimental design was to be able to represent the Cretaceous savanna biome with a combination of PFTs that exist in the ORCHIDEE framework. Modern savannas are particularly hard to represent in DGVMs (see for example Baudena et al. 2015, for a discussion) and so are the ones for the Cretaceous. Sewall et al. (2007) describe the Cretaceous savanna biome as a "dry, low understory with sparse broad leaved overstory" vegetation. The paleovegetation record mentions savannas composed of ferns (e.g. Coiffard et al., 2007) and "shrubby or herbaceous 'savannah-like' vegetation (without grasses)" (Bond and Scott 2010). We chose to represent this Cretaceous biome as a combination of 90 % C3-grassland and 10 % of tropical broad-leaved evergreen to best-fit Sewall et al. (2007) savanna description, i.e. to emulate a vegetation that is mostly of small height and that includes only little higher broadleaved components. We agree about uncertainties in this choice to translate paleobiomes into a combination of modern PFTs, but we think that is the best choice to make in the current model configuration. To be clearer about this choice, we add in the table caption: " Because savannas are particularly hard to represent in DGVMs (e.g Baudena et al. 2015), the savanna biome described by Sewall and al. 2007 as "dry, low understory with sparse broad leaved overstory" vegetation is represented by a combination of 90 % C3-grassland and 10 % of tropical broad-leaved evergreen. This choice emulates a vegetation that is mostly of small height and that includes only little higher broadleaved components, similar to the savanna description given by Sewall et al. (2007).

*In addition, I find it a bit hard to attribute the different kinds of greens in the map to the corresponding fields in the color bar, although I am not color-blind (but: one and the same hue in the map may look subjectively darker or lighter depending on the color hue next to it). Could you use more colors (the online version will be color-based)? Color-blind-friendly palettes can be found, for example, here: https://thenode.biologists.com/data-visualization-with-flying-colors/research/*

Thank you for your advice, we have changed the color palette of Figure 3. Specifically, as for the other figures, we used the perceptually uniform, colour-blind-friendly colourmaps developed and published by F. Crameri (https://www.fabiocrameri.ch/colourmaps/).

[Figure]

Vegetation distribution prescribed in the Aptian simulation configuration: map showing the dominant PFT for each grid cell. For all experiments, this distribution is fixed through time.

*l. 187/188 "We set the solar radiation at 99 % of the current value and the orbital parameters as today" reference/justification for that?*

The reference is Gough (1981) and is now added.

*l. 207 (ANGIO, Table 1): I'd call that the reference control experiment.*

Indeed, we add the term "reference" accordingly.

*l. 210-213: relationship between $g_s/g_{anat}^{max}$: I'm not so sure about that assumption. I understand $g_{anat}^{max}$ to be the upper constraint on operational $g_s$, but why should the ratio between $g_s$ and $g_{anat}^{max}$ be constant? It ought to vary on a daily time scale depending on the environmental factors that drive assimilation and transpiration, whereas $g_{anat}^{max}$ on short time scales remains a constant?*

Thanks for the remark, we were here considering the average $g_s$ value over a long time period as obviously $g_s$ vary in a short time. We corrected it in the text.

*l. 213: "the experiments were run for 60 years": I suppose you conducted equilibrium simulation runs. What about spin-up duration prior to the 60-year simulation? It must likely take longer than 60 years for biomass and PFT distribution to fully develop and equilibrate?*

For the purpose of the paper which focuses on transpiration and GPP, 60 years is more than enough to equilibrate, considering we do not use the DGVM. Although biomass would need more time to equilibrate. Equilibrium for GPP and transpiration - which are central to this study - is reached very quickly, 1-10 years after the beginning of the simulation.

*One general methodological question: What's the native temporal resolution of your output variables – daily, monthly sums/means, annual sums/means).*

The ORCHIDEE model simulates the water and energy exchanges between the continental surface and the atmosphere with a time step of 30 min.

***Results****:*

*I suggest to give subtitles to the different parts in the results section, e.g., 3.1 Stomatal conductance, 3.2 LAI and vegetation cover, 3.3 Canopy conductance, 3.4 Transpiration, 3.5 Water use efficiency (WUE); to make it easier to find the respective sections. You have structured it accordingly already, so just add the section titles for easier orientation.*

This is a very good suggestion, thank you. The manuscript has been modified accordingly.

*l. 222: "… and the C3 grass PFTs…": Again: I'm pretty sure there should be no C3-grass PFT during that time of the Cretaceous. 'Weedy' angiosperms maybe (according to William Bond's book on Open Ecosystems), but no C3-grass dominated biomes. Weedy angiosperms would not be grasses, but herbaceous non-woody life history forms.*

We fully agree about the absence of C3-grassland during the Cretaceous. We refer to the above explanations regarding our choices to represent Cretaceous savannas.

*l. 230/231: Logically makes sense within the assumptions of the framework - higher photosynthetic efficiency allows earlier closure of stomata with respect to the carbon aspect of stomatal regulation, because need for $CO_2$ is satisfied earlier, that is, with less open stomata.*

*General question: Do stomata generally in all plants open less fully under high $pCO_2$, or only when water becomes limiting? I imagine there may be different strategy types concerning this – for example, do swamp plants that never experience water stress open their stomata only to the degree required to take up the amount of $CO_2$ required to support the maximum rate that can be processed in the Calvin cycle, or do they simply not care, because transpirational water loss is never a constraint?*

This is obviously a very interesting question. In a generic photosynthesis model as Farquhar model (1980), used in ORCHIDEE, it assumes that stomatal aperture will always be optimized to minimize the water loss and then indeed will only go to the limit given by the Calvin cycle. But it is not guaranteed for plants that have never experienced water limitation. We know for instance that there are different responses of vegetation to hydric stress between isohydric or anisohydric plants. But here, it is a different strategy to

water limit: try to keep water at expense of productivity or try to maintain the productivity with the risk of hydraulic failure. In the case of $CO_2$ response there is no direct gain to keep stomata open. However for swamp plants, for instance, there is no selective pressure on this process and we can imagine that in such a context, some plants can have low stomatal response to $CO_2$.

*l. 240 LAI unit question: Is per $m^2$ of ground in reference to the actual vegetation-covered ground of a grid-cell (i.e., averaged across all PFTs), averaged across the canopy cover of the different PFTs, or the total grid cell area (i.e., potentially including bare-ground)? It would potentially be interesting to also see how much the bare ground area changes between the different factorial combinations, as bare ground proportion and vegetation density also indicate how well vegetation is performing.*

LAI is expressed per $m^2$ of ground of a grid-cell (i.e., surface averaged across all PFTs including bare soil). In other words, LAI is the weighted average of leaf area index of each PFT on the area they effectively occupy. We suggest writing ground$_{CELL}$ when we talk about the weighted mean over each PFT on the grid cell surface. We now mention that in the caption of Fig.5.

There is a prescribed fraction of bare soil that is fixed with the vegetation map over the 60-years of simulation (and is represented in the figure below).

[Figure]

Figure showing the prescribed fraction of ground (unitless) as bare soil during the 60 years of simulation.

However, when vegetation collapses in the sensitivity experiments, the actual bare soil fraction increases (i.e the actual surface that is not under vegetation) and depends dynamically on the LAI. This effective bare soil coverage is showed in the figure below.

[Figure]

Figure showing the actual bare soil fraction (unitless) for a) ANGIO(1120), b) ANGIO(280), c) NOAN-GIOh(1120), d) NOANGIOh(280), e) NOANGIOp(1120), f) NOANGIOp(280), g) NOANGIOhp(1120) and h) NOANGIOhp(280).

*l. 241 (and other occurrences) "peculiar". I would not use the term "peculiar", as it has a judgmental connotation. It may seem peculiar compared to present day, but was normal back then.*

We suggest replacing any occurrence of "peculiar" by "specific".

*l. 244-246: That is to be expected at $CO_2$ concentrations exceeding 1000 ppm, i.e., concentrations that are likely beyond the discussed $CO_2$ saturation point of C3 photosynthesis.*

Indeed, that result can be expected with $pCO_2$ at 1120 ppm. Still, further studies should at the same time investigate intermediate values and extrema from the Cretaceous, i.e. values ranging from modern-like (400 ppm) to 2000 ppm to explore the non-linearity of the processes described here.

*Figures/maps: given how different the Cretaceous conditions are from present day, I would find it extremely useful to have access to additional supplementary maps that show the climatic boundary conditions, at least average annual mean temperature and annual precipitation. In particular because $V_{cmax}$ (at least in our DGVM) is also temperature-dependent, i.e., a modulation of the reference $V_{cmax}25$ with temperature, and a temperature optimum beyond which $V_{cmax}$ declines again due to increased competitory binding of $O_2$ by the Rubisco, and ultimately starting degeneration of enzymes at yet higher temperatures.*

We now add surface temperature and precipitation in the supplementary information section of the paper. We are presently working on a companion paper aiming at assessing the climate response to the parameterizations developed here.

*l. 264: Maybe emphasize more strongly and explicitly that, unlike $g_s$, $g_c$ is the combined result of changes in $g_s$ AND changes in vegetation biomass and leaf area, i.e., both have an influence that can either be reinforcing or counterbalancing. Personally, I find it more interesting to look at $g_s$ than $g_c$, because that's where the physiology kicks in, whereas the $g_c$ aspect is more obscure due to the vegetation dynamics effect.*

Thank you, we agree that differences between $g_s$ and $g_c$ need to be clearly presented. We had attempted to make this point by writing specific paragraphs describing $g_s$ and $g_c$, but we acknowledge there was room for improvement. We enhanced the paragraph describing $g_s$ response and also modified the text to emphasize that $g_c$ is a more complex combination of $g_s$ and LAI dynamics. However, in order to interpret LAI, GPP and transpiration at the global scale, we still must analyse $g_c$. The focus of the paper is not to use the individual scale but more to assess the water (via the transpiration rate) and the carbon (via GPP) fluxes at the canopy scale. Moreover, in line with the next paper on global atmosphere-vegetation interactions, we are working on, the physiological response of the entire canopy will help us more to interpret atmospheric perturbations than the leaf scale.

*Fig. 4 and 6: Are these the global-scale annual averages for each PFT, or what exactly is the reference basis? I would expect strong spatio-temporal variations here on short time scales, depending on soil water content/water stress, temperature, soil conditions, etc. How about showing error bars/standard deviations in addition to the means, or are these too large to show on this type of figure?*

[Figure]

Global scale leaf operating stomatal conductance to $H_2O$ $g_s$ (mol m$^{-2}$[leaf] s$^{-1}$) per PFT: (a) ANGIO(1120) (gray bar), NOANGIOh(1120) (diagonal hatched bar), NOANGIOp(1120) (dotted bar), and NOANGIOhp(1120) (diagonal hatched and dotted bar); (b) ANGIO(280) (gray bar), NOANGIOh(280) (diagonal hatched bar), NOANGIOp(280) (dotted bar), and NOANGIOhp(280) (diagonal hatched and dotted bar). The leaf stomatal conductance to $H_2O$ is the daylight average over the year and is weighted by unit of foliar surface per PFT. Standard deviations are red bars.

[Figure]

Global scale canopy operating stomatal conductance to $H_2O$, g$_c$ (mol m$^{-2}$[ground$_{PFT}$] s$^{-1}$) per PFT: (a) ANGIO(1120) (gray bar), NOANGIOh(1120) (diagonal hatched bar), NOANGIOp(1120) (dotted bar), and NOANGIOhp(1120) (diagonal hatched and dotted bar); (b) ANGIO(280) (gray bar), NOANGIOh(280) (diagonal hatched bar), NOANGIOp(280) (dotted bar), and NOANGIOhp(280) (diagonal hatched and dotted bar). The canopy stomatal conductance to $H_2O$ is the daylight average over the year and is weighted by the surface grid where each PFT is found. Standard deviations are red bars.

Yes, the figures 4 and 6 are global scale annual mean of $g_s$ (and $g_c$) of each PFT and weighted per m$^2$ of leaf (ground$_{PFT}$) where each PFT is effectively present. For the PFT scale, we suggest writing ground$_{PFT}$ to distinguish from ground$_{CELL}$. The figure caption is rephrased to be accurate. On the figures above, we have added standard deviation over each bar (red), calculated from the last 10 years annual averaged $g_s$ and $g_c$. Globally, they are very small because of little inter-annual variations for the last 10 years of simulations.

In absolute terms, some differences arise from the uptake of water and solar energy between the low latitude region and the mid to high latitude region (figures below). However, there are little changes when comparing two simulations (anomaly) for any specific region. As the focus of the paper is to investigate the differences between sensitivity tests and the reference ones, we choose to show only the global means. The figures below show $g_s$ and $g_c$ for the low latitudes (30°-30°N) and the mid-to high latitudes (30°N-90°N).

**Leaf operating stomatal conductance per region:**

[Figure]

Low latitude leaf operating stomatal conductance to H$_2$O $g_s$ (mol m$^{-2}$[leaf] s$^{-1}$) per PFT: (a) ANGIO(1120) (gray bar), NOANGIOh(1120) (diagonal hatched bar), NOANGIOp(1120) (dotted bar), and NOANGIOhp(1120) (diagonal hatched and dotted bar); (b) ANGIO(280) (gray bar), NOANGIOh(280) (diagonal hatched bar), NOANGIOp(280) (dotted bar), and NOANGIOhp(280) (diagonal hatched and dotted bar). The leaf stomatal conductance to H$_2$O is the daylight average over the year and is weighted by unit of foliar surface per PFT in the low latitudes (30°S to 30°N). Standard deviations are red bars.

[Figure]

Mid to high latitude leaf operating stomatal conductance to $H_2O$ $g_s$ (mol $m^{-2}$[leaf] $s^{-1}$) per PFT: (a) ANGIO(1120) (gray bar), NOANGIOh(1120) (diagonal hatched bar), NOANGIOp(1120) (dotted bar), and NOANGIOhp(1120) (diagonal hatched and dotted bar); (b) ANGIO(280) (gray bar), NOANGIOh(280) (diagonal hatched bar), NOANGIOp(280) (dotted bar), and NOANGIOhp(280) (diagonal hatched and dotted bar). The leaf stomatal conductance to $H_2O$ is the daylight average over the year and is weighted by unit of foliar surface per PFT in the mid to high latitudes (30°N to 90°N). Standard deviations are red bars.

**Canopy operating stomatal conductance per region:**

[Figure]

Low latitude canopy operating stomatal conductance to $H_2O$, $g_c$ (mol $m^{-2}$[ground$_{PFT}$] $s^{-1}$) per PFT: (a) ANGIO(1120) (gray bar), NOANGIOh(1120) (diagonal hatched bar), NOANGIOp(1120) (dotted bar), and NOANGIOhp(1120) (diagonal hatched and dotted bar); (b) ANGIO(280) (gray bar), NOANGIOh(280) (diagonal hatched bar), NOANGIOp(280) (dotted bar), and NOANGIOhp(280) (diagonal hatched and dotted bar). The canopy stomatal conductance to $H_2O$ is the daylight average over the year and is weighted by the surface grid where each PFT in the low latitudes (30°S to 30°N). Standard deviations are red bars.

[Figure]

Mid to high latitude canopy operating stomatal conductance to $H_2O$, $g_c$ (mol m$^{-2}$[ground$_{PFT}$] s$^{-1}$) per PFT: (a) ANGIO(1120) (gray bar), NOANGIOh(1120) (diagonal hatched bar), NOANGIOp(1120) (dotted bar), and NOANGIOhp(1120) (diagonal hatched and dotted bar); (b) ANGIO(280) (gray bar), NOANGIOh(280) (diagonal hatched bar), NOANGIOp(280) (dotted bar), and NOANGIOhp(280) (diagonal hatched and dotted bar). The canopy stomatal conductance to $H_2O$ is the daylight average over the year and is weighted by the surface grid where each PFT in the low latitudes (30°N to 90°N). Standard deviations are red bars.

*In addition: It's a bit surprising to see that there are reductions for the NOANGIOp and NOANGIOhp scenarios for the temperature needleleaf evergreen vegetation, as these gymnosperms were not manipulated with respect to stomatal conductance and/or photosynthetic capacity. These effects therefore (given no mistakes were made) must be indirect. Is it because the global area covered by this PFT and its spatial distribution are different due to the change in angiosperm performance and distribution? I'd expect that $g_c$ (and $g_s$) are also very variable spatially as well as temporally. In any case, this point needs to be discussed/explained.*

The global area covered by gymnosperm PFTs and its spatial distribution are not different because that is fixed for each time step. We explain the decrease of gymnosperm $g_c$ from feedback from the atmosphere-vegetation coupling that ultimately altered soil moisture stress and stomatal conductance. Indeed, soil moisture stress for transpiration increases (positive anomaly) where gymnosperm is prescribed for all the perturbed simulations with the proto-angiosperm vegetation prescribed because of less precipitations (figures below.

[Figure]

Soil moisture stress for transpiration of gymnosperm PFT (unitless) for (a) ANGIO(1120) and (b) AN-GIO(280), anomalies of annual soil moisture stress for transpiration of gymnosperms (*10 - unitless) for (c) NOANGIOh(1120) vs ANGIO(1120), (d) NOANGIOh(280) vs ANGIO(280), (e) NOANGIOp(1120) vs ANGIO(1120), (f) NOANGIOp(280) vs ANGIO(280), (g) NOANGIOhp(1120) vs ANGIO(1120) and (h) NOANGIOhp(280) vs ANGIO(280). The t-test 95 % confidence level anomalies are given by dots.

[Figure]

Annual mean precipitation rates (mm d$^{-1}$) for (a) ANGIO(1120) and (b) ANGIO(280), anomalies of annual mean precipitation rates (mm d$^{-1}$) for (c) NOANGIOh(1120) vs ANGIO(1120), (d) NOANGIOh(280) vs ANGIO(280), (e) NOANGIOp(1120) vs ANGIO(1120), (f) NOANGIOp(280) vs ANGIO(280), (g) NOANGIOhp(1120) vs ANGIO(1120) and (h) NOANGIOhp(280) vs ANGIO(280). The t-test 95 % confidence level anomalies are given by dots.

*l. 266: "... by surface unit of ground...": surface unit ground attributed to the PFT? Surface unit ground actually covered by crown area of the PFT?*

Yes, by surface unit of ground attributed to each PFT. We rephrase the sentence as the figure caption.

*l. 275/276: "Indeed, at the canopy scale, the closing of the stomata at high $pCO_2$ is compensated by the decrease in LAI at low $pCO_2$." I'm not sure I understand that sentence. Do you mean that higher $g_c$ at low $pCO_2$ just makes up the LAI loss, so that the lower $g_c$ at high $pCO_2$ x higher LAI is just about balanced against each other?*

We agree, the sentence should be clearer. We replace the original sentence by: "Indeed, at the canopy scale, at low $pCO_2$, higher $g_s$ just makes up the LAI loss. Conversely, at high $pCO_2$, lower $g_s$ is compensated by higher LAI, ultimately making angiosperm $g_c$ being almost identical for the two $pCO_2$ scenarios."

*l. 284: GPP, Fig S2: GPP is in units "per $^2$". Is this per $^2$ of leaf area, per $^2$ ground area covered by vegetation, per $^2$ canopy area, per $m^2$ of grid cell area...?*

GPP is weighted by $m^2$ of ground effectively covered by each PFT. So per $m^2$ ground$_{CELL}$ area covered by vegetation.

*I'd find it easier to see the differences between the scenarios that are solely due to the physiological manipulations if GPP were normed to unit leaf biomass, or unit leaf area. Could you provide such a figure in addition to Fig S2?*

The figure above shows the annual mean GPP/LAI. GPP/LAI is the carbon assimilation per leaf area unit ($kgC\ y^{-1}\ m^{-2}$[leaf]). It is higher at high $pCO_2$ than at low $pCO_2$ for the reference experiments because of the fertilization effect. However, the very low LAI values in most sensitivity experiments (Fig. 5) makes it difficult to interpret a GPP/LAI ratio, which is very high. We develop the reason why we show carbon (GPP) and water (transpiration) fluxes at the canopy scale thereafter.

[Figure]

Annual mean effective GPP or carbon assimilation (kgC y$^{-1}$ m$^{-2}$[leaf]) for a) ANGIO(1120), b) ANGIO(280), c) NOANGIOh(1120), d) NOANGIOh(280), e) NOANGIOp(1120), f) NOANGIOp(280), g) NOANGIOhp(1120) and h) NOANGIOhp(280). The annual mean GPP is the weighted average of gross primary productivity of each PFT on the area they effectively occupy.

*l. 286: "... and have only an indirect effect on GPP...": I'd say that's a rather direct effect, not indirect, as it directly affects assimilation due to C-limitation of assimilation?*

What we want to discuss here is the different magnitude of the different perturbed experiments whatever the $pCO_2$ case. In fact, the GPP response is different depending on the parameter affected ($g_s$ directly via fcpl*$\alpha$) or (A directly via $V_{cmax}$*$\beta$) or both. Decreasing $V_{cmax}$ has a direct effect on GPP because GPP is the integral of A over the canopy (loops over the LAI levels) whereas decreasing fcpl has only an impact on GPP because of the coupling between $g_s$ and A, that is the reason why we use the word "indirect". Thus, we see that the change in GPP is lower for NOANGIOh experiments than for NOANGIOp experiments (compare Fig S2c to S2e or S2d to S2f).

*l. 293: "... that depends on air humidity...": I find it physically more accurate to say it depends on the vapor pressure deficit.*

We rephrase it accordingly.

*l. 294/295: "... the capacity of plants to transpire, driven by the canopy conductance.": In detail, it's a combination of coupled conductance terms (or maybe not in ORCHIDEE?) - stomatal conductance, leaf boundary layer conductance, and canopy boundary layer conductance?*

Yes, we agree (cf comment above). We mention the different conductances in the model formulation and in the summary.

*l. 301/202: "Parameterizing the vegetation without the modern angiosperm hydraulic and photosynthetic capacities systematically leads to lower transpiration rates (Fig. 7).": In absolute terms, that's no surprise, given that there is less leaf area available to transpire water. However, does this also hold in relative terms, i.e., when looking at the amount of water transpired per unit leaf area or per unit leaf biomass, per year?*

We agree, it is not surprising. But if transpiration responds to the different parameterizations, it is also a way to confirm our experimental design.

We cannot look at the amount of water transpired per unit leaf area, since dividing transpiration rate by LAI would involve that each canopy layer has the same weight on transpiration (and GPP, cf question above). However, leaves at the top of the canopy do not develop and transpire to the same extent as the low layer. In our land-surface model, the transpiration rate and the GPP are not calculated by leaf unit, which makes it difficult to access the notion of "effective transpiration" (mm m$^{-2}$[leaf] d$^{-1}$) and "effective GPP" (gC m$^{-2}$[leaf]). The model is built to give the carbon and water fluxes at the canopy scale. The figures below (transpiration/LAI) give an idea of the amount of water that is transpired per unit of leaf. As expected, effective transpiration is higher at low $pCO_2$ than at high $pCO_2$ because of the opening of stomata. But if the values are so high for the sensitivity experiments at low $pCO_2$ that is because the LAI is near zero. The figures give the idea that there is a lot of transpiration for NOANGIOp(280) and NOANGIOhp(280) while it is near zero (cf Fig. 7). At high $pCO_2$, this is not all the leaves that transpire but mostly those at the top of the canopy. Thus, this figure does not really make sense.

[Figure]

Annual mean effective transpiration rate (mm m$^{-2}$[leaf] d$^{-1}$) for (a) ANGIO(1120) and (b) AN-GIO(280), anomalies of annual mean transpiration rate (mm m$^{-2}$[leaf] d$^{-1}$) for (c) NOANGIOh(1120), (d) NOANGIOh(280), (e) NOANGIOp(1120), (f) NOANGIOp(280), (g) NOANGIOhp(1120) and (h) NOAN-GIOhp(280).

The physiological behavior of the vegetation is represented by the stomatal conductance at the leaf scale $g_s$. The leaf stomatal conductance (figure 4) is thus the good proxy of associated changes in the effective transpiration and GPP at the leaf level. The idea of the paper is to build step by step the rationale from the parameterization at the leaf scale, to the energy and water fluxes at the canopy scale in order to assess the transpiration and photosynthesis changes at the global scale. We clarify that point at the end of the introduction and the plan is announced as we look at the stomatal conductance from the leaf scale to the canopy scale, in order to finally assess the water and carbon exchanges between the land-surface and the atmosphere.

*l. 306/307: "Transpiration also significantly drops when photosynthetic capacity alone is reduced...": This is most likely due to the decrease in leaf biomass/LAI, right? Again, to allow focusing on the physiological reactions without the confounding effects resulting from differences in leaf biomass and leaf area, norming transpiration to either state variable would be helpful.*

Transpiration is calculated from the evaporative demand and the canopy stomatal conductance, that arises from a combination of LAI and leaf stomatal conductance (Eq. (5)). Indeed, transpiration variations (Fig. 7) are similar to that of LAI (Fig. 5), but are also impacted by the decrease of $g_s$. Such a decrease is significant (Fig. 4) as we estimate it as ⅓ of the reference experiment. So, both $g_s$ and LAI drive the response of plant transpiration.

*l. 315: Transpiration anomalies: Would be interesting to also have information on the latent heat flux deficit and latent heat flux deficit changes, i.e., the ratio between actual transpiration and potential evapotranspiration, as this is an indicator for water stress.*

We are not sure that transpiration/evapotranspiration would provide information about the water stress, as it would only provide a fraction of the latent heat flux terms, missing evaporation of water on the leaves and on the soil. Also, figure S2 depicts directly the soil water stress computed by the model.

[Figure]

Annual mean transpiration rate (mm d$^{-1}$) on potential evapotranspiration rate (mm d$^{-1}$) for (a) AN-GIO(1120), (b) ANGIO(280), (c) NOANGIOh(1120), (d) NOANGIOh(280), (e) NOANGIOp(1120), (f) NOANGIOp(280), (g) NOANGIOhp(1120) and (h) NOANGIOhp(280).

*l. 319: "... while arid belt regions are less sensitive to any change in $g_c$.": I guess transpiration is already constrained by water shortage there, so the upper stomatal conductance limit is not the limiting factor anyway. If that is true, maybe explicitly mention that as an explanation?*

That is exactly what we meant. We add the explanation accordingly.

*l. 334/335: "... with a 1.7 gC kgH2O$^{-1}$ (+30 %) increase compared to ANGIO (Fig. 8a and c)." Show anomaly-to-control figs in supplementary material?*

We added figures of WUE anomaly-to-control in the supplementary material.

Discussion:

*l. 362/363: Hence, a lower maximal stomatal conductance at high $pCO_2$ appears as an advantage compared to modern angiosperm because of a better optimization of carbon uptake over water loss.": Here, I'm struggling a bit conceptually. Yes, a lower maximal stomatal conductance at high $pCO_2$ is an advantage in terms of water loss, but does not impact GPP, because Ci is not limiting even under reduced stomatal conductance. That much is clear.*

What we would like to say is that for a constant GPP, lowering the maximal stomatal conductance at high $pCO_2$ reduces water loss, so the ratio of carbon uptake over water loss is higher.

*However, does it necessarily have to be the structural conductance that needs to be lower to achieve this effect? Would it not also be possible for modern angiosperm plants with high structural conductance to achieve the same effect, simply by keeping the stomata only as much open as is required to satisfy the maximum carbon demand required to just make Ci not limiting any more? This goes into the direction of optimality in "operational" stomatal regulation, in the sense of which environmental factor is limiting under given circumstances. Given water is not limiting, it should be in a plant's interest to open the stomata just wide enough to acquire enough carbon to satisfy the demand from the photosynthesis, i.e., to take up the amount of carbon that can instantaneously be processed in the Calvin cycle, so that Ci is not limiting. Under these circumstances, carbon drives stomatal regulation. If water becomes limiting, the plant may have to balance maximum water loss against carbon gain, by closing the stomata enough to, maximally, only allow the amount of water to be transpired that can be provided through the SPA-continuum. In this case, water availability should drive stomatal regulation, which implies that Ci can become limiting for photosynthesis due to the water constraint, if Ci would demand a stomatal opening exceeding the one allowed by water availability.*

Increasing leaf stomatal conductance by ramification of the vein network leads to a cost in terms of material investment (Chapin et al., 2011; Fiorin et al., 2016; Beerling et Franks, 2010). To optimise the carbon uptake over water loss, plants must coordinate the production of veins with tissues responsible for photosynthetic gas exchange (Brodribb et al., 2007). Unless a high stomatal density is matched by a high vein density, stomata will be forced to remain partially closed (Dow and Bergmann 2014). Moreover, there is an optimal range of $g_s/g_{anat}^{max}$ among the plants (Dow and Bergmann 2014), which is found to be constant on a long time scale (Dow et al., 2014). Thus, it seems difficult to imagine that plants strongly increase the degree of opening of

its stomata without increasing its structural conductance.

*Stomatal closure under water stress will also lead to a reduction in transpiration, i.e., latent heat flux. To fulfill the energy balance, this entails an increase in sensible heat flux associated with an increase in leaf temperature, which, via temperature dependence of $V_{cmax}$, affects assimilation and therefore Ci. To solve this, an approach is required that simultaneously solves for stomatal conductance, assimilation, leaf temperature and transpiration.*

Our land surface model is already complex, with stomatal conductance and carbon assimilation coupled together. A simplification comes from the single energy budget, which translates into a single surface temperature within the canopy. While $V_{cmax}$ and other transpiration- and photosynthesis-related parameters are sensitive to leaf temperature, this is for sure a simplification. Recent effort has been done to develop a multi-canopy-layer energy budget in ORCHIDEE (Ryder et al. 2016), which would benefit in the future to the ORCHIDEE version coupled to the LMDz atmospheric model.

*l. 367: "by decreasing modern $V_{cmax}$": By what mechanism would such a reduction of modern $V_{cmax}$ have been accomplished - maybe briefly discuss. Rubisco binding efficiency per unit Rubisco molecule should have been the same as today, unless there was a different version of Rubisco back then that was less efficient. So likely a reduction would have been associated with lower concentrations of Rubisco in plant cells?*

Here we were talking about reduction of $V_{cmax}$ done in previous modeling approaches (Boyce et Lee, 2010; Lee et Boyce, 2010) not in reality and it was then applied just by decreasing the $V_{cmax}/V_{jmax}$ value. It is what we did in NOANGIOp experiments to show that applying only change on $V_{cmax}$ was not realistic. However, we showed that applying a co-limitation of $V_{cmax}/V_{jmax}$ and $g_s$ under high $pCO_2$ is probably the most realistic solution, even if we cannot have evidence on change in rubisco concentration in the paleo records. As we explain in the text, this is not related to a change in Rubisco efficiency which had, indeed, no reason to change, but it is related to the energetic cost needed to maintain rubisco and acquire nitrogen. So to optimize the nitrogen use efficiency, plants adapt their rubisco concentration to be at the level of co-limitation between $V_c$ and $V_j$ as stated by the coordination theory.

*l. 384/385 "They argue that modern angiosperm trees have 2 times higher stomatal conductance sensitivity response to driving factors than gymnosperm trees": How does $V_{cmax}$ of modern gymnosperm trees compare to that of modern angiosperm trees? Is it relatively lower as well, and if, then by how much?*

Yes, indeed the $V_{cmax}$ of gymnosperms is lower than for angiosperms. For instance the value in ORCHIDEE of $V_{cmax}$ is around 50 for angiosperms trees and around 35 for coniferes. However, this is not only related to the difference between gymnosperms and angiosperms but also to the difference in specific leaf area and leaf life span (leaf economic spectrum).

*l. 394/395 "However, lowering only the hydraulic capacity while keeping the high $V_{cmax}$ as in the modern vegetation induces a nitrogen cost.": This nitrogen cost would also exist under high hydraulic conductivity. It might only have a lower impact if the entire C-balance of the plant is improved due to the improved stomatal conductivity, i.e., under circumstances where stomatal conductivity is limiting to Ci.*

Indeed our sentence was not clear enough. Our objective was to say that, under high $pCO_2$, because of increasing limitations of $V_j$, the plant cannot benefit from a high $V_{cmax}$. Hence the actual photosynthesis can be maintained with a lower $V_{cmax}$. Meanwhile, maintaining a high $V_{cmax}$ implies a nitrogen acquisition and protein maintenance costs. So, to optimize the ratio of GPP to respiration, we expect a decrease of $V_{cmax}$ (coordination theory). So we rewrite the sentence accordingly: "However, lowering only the hydraulic capacity while keeping the high $V_{cmax}$ as in the modern vegetation is probably not realistic. Indeed, under high $pCO_2$, because of increasing limitation of rubisco regeneration, the actual photosynthesis can be maintained with a smaller $V_{cmax}$. As maintained, a given level of $V_{cmax}$ is associated with a respiration cost (for nitrogen acquisition and protein maintenance). So to optimize it carbon gain, we expect the plant to also have a lower $V_{cmax}$".

*l. 395 "Although our model does not represent the nitrogen cycle...": You should mention that already in the model description, in particular because there is an ORCHIDEE version that has a nitrogen cycle (ORCHIDEE-CN).*

Corrected.

*l. 404/405: Do not most of the modern DGVMs and land surface models do that anyway for quite a long time now?*

Yes indeed ! We rephrased to: "The study also shows that paleovegetation characteristics can be better represented by parameterizing models fully describing the coupling between stomatal conductance and plant productivity from leaf to the canopy scale."

*l. 408/409: "... leaf stomatal conductance is only 3-time lower than the reference": That is actually one of the points that merits a bit more explanation, because it is somewhat unexpected/counterituitive. My guess would be that the stomatal conductance in the control scenario is not always and everywhere at its maximum, therefore the relative reduction compared to a value that was already constrained in the control scenario (e.g., due to light limitation on photosynthesis or water limitation effects on stomatal conductance) is less than the full possible range?*

Yes, this is exactly the reason ! Indeed, because the factor is applied only to the anatomical maximum conductance, it does not fully propagate to the operational conductance which depends on multiple external forcings and which is lower than the maximum conductance. This is what we stated in the text: "Furthermore, we show that decreasing hydraulic and/or photosynthetic capacities does not coincide with a decrease of the leaf operational stomatal conductance to the same extent. Indeed, accounting for a decrease by a factor of 5, given by the maximal bound of the range expected from the maximal anatomic stomatal conductance, leaf stomatal conductance is only 3-time".

We try to correct it to be more clear as: "The stomatal conductance in the reference experiments (ANGIO) is not always and everywhere at its maximum due to light limitation or water limitation effects on photosynthesis. That is because the $\alpha$ and $\beta$ factors are applied only to the model parameters that correspond to the

anatomical maximum conductance, and they do not fully propagate for the operational conductance, which depends on multiple external forcing."

l. 411/412: "we suggest to explicitly represent changed in hydraulic and photosynthetic capacities.": I'd suggest that the gold standard to aim for should be a fully coupled SPA-water transport continuum based on coupled resistances that is linked to assimilation/assimilatory demands and leaf temperature regulation, constrained by energy balance requirements (sensible heat flux vs. latent heat flux).

We agree and added a sentence in the perspective that further works will need to include a fully coupled SPA-water transport continuum. Especially as for leaf structure, there is also some paleo data on structural changes in plant vessels.

l. 418/419 "… and water availability in the soil…": In addition also to VPD (potential gradient between near-saturated conditions in leaf-intercellular air space vs. leaf-exterior conditions).

Yes indeed, VPD was missing here ! It is corrected.

l. 450 "compilation and spatialization of the Aptian paleobotanical records (Sewall et al., 2007)": Looking at Sewall et al, plus other literature resources, there should be no C3-grass biomes existing during the Aptian, because C3 grasses likely did not evolve in abundance before the Maastrichtian, and their rise to dominance in open biomes did not happen before the Tertiary.

Thanks again for the comment, we discussed that point in the replies above.

l. 456/457 "by allowing PFTs to spatially settle in regions where the simulated climate is the most appropriate": It would be even better to go beyond PFT-based dynamic vegetation modeling, for example by directly using DGVMs as land surface scheme that are trait-based and therefore allow for direct inclusion of trait-based modelling approaches.

We fully agree, but such novel approaches under which all the DGVM community are progressively converging are still, at least for what we know, under development.

l. 458: studies on Paleozoic vegetation transitions: the Cretaceous period that you are interested in is part of the Mesozoic, not the Paleozoic.

Yes, indeed. We talk about Paleozoic as it is another example of paleo vegetation transition.

l. 461/462 Aside from Vuichard et al., 2019, also cite Soenke Zaehle's older work on OC-N here (Zaehle and Friend 2010 ; Zaehle, Friend, et al. 2010? Or has that by now become obsolete for the development of the N-cycle branch of ORCHIDEE?

The work of Zaehle et al., (2010) is still relevant. References are now added.

*l. 462/463 "provided that good constraints can be obtained regarding the C:N ratio of Cretaceous vegetation and soils" => that will very likely always be a big source of uncertainty. Also, I'm not sure when symbiotic N-fixation evolved and whether it would already have mattered during the Aptian, i.e., how abundant symbiotic N-fixers would have been if they already existed.*

Yes indeed, it was the objective of the sentence to specify that a model including nitrogen cycle would be important (for instance to better constrain the change in $V_{cmax}$ that was forced in the study) but on the other hand is very difficult to constrain for the Cretaceous.

*l. 473/474 Yes, this is also where I see the future. Stomatal conductance, the way I perceive it, is a two-way road that links C-gain against associated water loss. Depending on which factor is more limiting, either the one or the other will be driving stomatal regulation.*

We share this view.

*l. 478/479 "It emphasizes the need, in the future, to improve the parameterization of stomatal conductance in global mode by explicitly modelling both structural and dynamic conductance." => Do you have a more concrete suggestion on how this should be implemented, and an estimate how much different it would make simulation results?*

Some attempts have been made to define such a fully explicit model even if still not complete. For instance, Dow and al. (2014) developed such a model on 6 angiosperm individuals with measurements under controlled conditions. Without an adaptation to PFTs in a land-surface model and to the global scale, it is very difficult to estimate how it could change our results. However, it would allow to associate more directly parameters to observations.

***Conclusions:***

*l. 484 "with an ecophysiological model based on angiosperm fossil records." => "motivated by an ecophysiological model" is more appropriate. You are mimicking some of the constraints of the ecophysiological model, but do not fully implement it.*

We agree and modified the text accordingly.

*l. 503 "even at an expense of possible increasing water loss" => The water loss aspect should be treated/discussed in a more differentiated manner: where and when water is not limiting, it is a secondary problem. Where and when water IS limiting, the question is whether an increased structural conductance actually really results in an increased water loss. I have some doubts about that, because under the aspect of optimality, it is likely that plants, even with a higher structural conductance, would nonetheless close the stomata enough to limit water loss to the amount that can be provided through the SPA-continuum. In that case, one would not necessarily expect drastically increased water loss although it would be potentially possible. So in both cases, the problem of increased water loss should be secondary.*

We agree. Here we mention water loss to allow the reader to make the link with water balance in the plant, which is of interest. Moreover, several studies suggest an extra cost for plants which have a higher stomatal conductance without increasing the structural conductance (Chapin et al., 2002 ; Fiorin et al., 2015 ; Beerling et Franks, 2010 ; Dow et Bergmann, 2014 ; Dow et al., 2014).

*l. 511/512 "Furthermorel allowing dynamic vegetation would be an important future refinement of this research to model feedbacks between vegetation and climate." => In addition, also moving beyond PFT-based approaches towards truly trait-based approaches, e.g., such as the ones pursued by Scheiter et al. with the aDGVM2 model, or the JEDI model developed by Pavlick and colleagues. It would allow a more direct accounting for changes in functional traits, as well as a dynamic evolutionary selection for those trait strategies that are competitively successful under given environmental conditions.*

Thanks, we added references to aDGVM2 and JEDI in the text: "Also, replacing PFT morpho-physiological traits by species-specific traits (Kattge et al. 2011) as it has already been done for the aDGVM2 model (Scheiter et al., 2013), the LPJmL-FIT model (Sakschewski et al. 2015) or the JEDI model (Pavlick et al. 2013), allows plant communities to be assembled based on how plants with different trait combinations perform under a given set of environmental conditions. In this way, changes in functional traits would be more directly taken into account."

**Data availability:**

*How about code availability? In the interest of Reproducible Science, making source code required to reproduce the results available is about to become standard. Is the source code used for the simulations in this study available, or can it be made available (e.g., in an online repository such as Github)? If not, please state why it is not possible.*

This is an important point. The LMDZ code is publicly available as part of the IPSL-CM5A2 earth system model, that has been made available by Sepulchre et al. (2020). The code can be retrieved through svn, with the following command lines:
 svn co http://forge.ipsl.jussieu.fr/igcmg/svn/modipsl/branches/publications/IPSLCM5A2.1_11192019/ (last access: 25 June 2020; Gatthas, 2020)
See also: https://gmd.copernicus.org/articles/13/3011/2020/#section11
The ORCHIDEE code, that has been modified for this study, is available as well through svn: svn co http://forge.ipsl.jussieu.fr/igcmg/svn/modipsl/branches/publications/ORCHIDEE_IPSLCM5A2.1.r5307. The login/password combination requested at first use to download the ORCHIDEE component is anonymous/anonymous.

**Suggested minor corrections and changes:**

*l. 8 generates => generate*
*l. 61 links => link*
*l. 171 lower => smaller*
*l. 187 orbitals => orbital*
*l. 101 suggested => suggesting*

*l. 205 modern-like => modern-type*

*l. 260 which plays at => which plays a role at*

*l. 280: "LAI changes described earlier act as a feedback between $g_s$ and $g_s$": I'd rather say: "LAI changes described earlier modulate the relationship between changes in $g_s$ and $g_s$."*

*l. 301 angiosperms => angiosperm*

*l. 315 come from => result from*

*l. 330 plants adaptation => plant adaptation*

*l. 334: NAONGIOh => NOANGIOh*

*l. 358 "... to account for the decrease of maximal stomatal conductance": ... to account for the lower maximum stomatal conductance..." (decrease has a temporal connotation, for a change from first high to then low).*

*l. 360 decreasing fcpl => lower fcpl*

*l. 363 modern angiosperm => modern angiosperms*

*l. 364 "... because of the positive feedbacks of the LAI on the canopy stomatal conductance," => "because a reduced LAI entails a reduction of canopy stomatal conduction..."*

*l. 387 "..., that remains enough to sustain the LAI...": "that remains sufficient to sustain LAI values close to those of the control scenario..."*

*l. 399 consistent with previous study => consistent with a previous study*

*l. 402 while sustaining high productivity => that allowed nonetheless to sustain a high*

*productivity l. 403 as in the modern vegetation => as found in the modern vegetation*

*l. 403 preserve => maintain*

*l. 408 3-time => either "3-times" or "3-fold"*

*l. 425 basals angiosperms => early angiosperms*

*l. 425 was as low as the other plant types => was as low as that of other plant types*

*l. 427 At that time => for this time*

*l. 427 "... we confirm the hypothesis..." => "... our results support the hypothesis..."*

*l. 427 "... evolved towards leaves more and more densely irrigated together with..." => "... evolved towards leaves with increasing vein density combined with a..."*

*l. 428 increasingly stomatal conductance => increasing stomatal conductance*

*l. 429 Among others => Among other factors*

*l. 430 dominating the vegetation of the period to colonize => dominating the vegetation of the period that enabled them to colonize*

*l. 431 with that of Franks => with those of Franks*

*l. 432 "periods with high pCO$_2$ strengthen GPP, meanwhile a potential decrease of transpiration rate by the closing of the stomata" => "periods with high pCO$_2$ enhanced GPP while simultaneously allowing a reduction of transpirational water losses due to reduced stomatal conductance"*

*l. 438/439 invested increasingly energy => invested increasingly more energy*

*l. 440 in densely water transport networks => in dense water transport networks*

*l. 458 studies about Paleozoic vegetation transitions => studies on Paleozoic vegetation transitions*

*l. 490 which is not recorded in the fossil => which is in contradiction to the fossil record*

*l. 492 All the results taken together => All the results in combination*

*l. 495 "Therefore, the combining decrease of hydraulic and photosynthetic capacities..." => "Therefore, a combination of lower-than-modern hydraulic and photosynthetic capacities...". "Decrease" implies a temporal*

*dynamic from high towards low*

*l. 498/499 This result is also consistent => Our results are also consistent*

*l. 500/501 "... was adapted to high $pCO_2$ by sustaining productivity and a high WUE." => "... was adapted to high $pCO_2$, where the combination of both physiological constraints nonetheless allowed high productivity and WUE."*

*l. 501 "it was not adapted to lower $pCO_2$ as GPP collapses" => "it would not have been able to exist under low $pCO_2$, where we simulated a collapse of GPP under such physiological parameter configuration."*

We thank you for all the minor suggestions you have made. We rephrase sentences accordingly.

**References**

Baudena, Mara et al. (2015). "Forests, savannas, and grasslands: bridging the knowledge gap between ecology and Dynamic Global Vegetation Models". In: *Biogeosciences* 12.6, pp. 1833–1848. DOI: https://doi.org/10.5194/bg-12-1833-2015.

Beerling, David J and Peter J Franks (2010). "The hidden cost of transpiration". In: *Nature* 464.7288, pp. 495–496. DOI: https://doi-org.insu.bib.cnrs.fr/10.1038/464495a.

Bond, William J and Andrew C Scott (2010). "Fire and the spread of flowering plants in the Cretaceous". In: *New Phytologist* 188.4, pp. 1137–1150. DOI: https://doi.org/10.1111/j.1469-8137.2010.03418.x.

Boyce, C Kevin and Jung-Eun Lee (2010). "An exceptional role for flowering plant physiology in the expansion of tropical rainforests and biodiversity". In: *P. Roy. Soc. B-Biol. Sci.* 277.1699, pp. 3437–3443. DOI: https://doi.org/10.1098/rspb.2010.0485.

Brodribb, Tim J and Taylor S Feild (2010). "Leaf hydraulic evolution led a surge in leaf photosynthetic capacity during early angiosperm diversification". In: *Ecol. Lett.* 13.2, pp. 175–183. DOI: https://doi.org/10.1111/j.1461-0248.2009.01410.x.

Brodribb, Tim J, Taylor S Feild, and Gregory J Jordan (2007). "Leaf maximum photosynthetic rate and venation are linked by hydraulics". In: *Plant Physiol.* 144.4, pp. 1890–1898. DOI: https://doi.org/10.1104/pp.107.101352.

Brundrett, Mark C (2002). "Coevolution of roots and mycorrhizas of land plants". In: *New Phytol.* 154.2, pp. 275–304. DOI: https://doi.org/10.1046/j.1469-8137.2002.00397.x.

Chapin III, F Stuart, Pamela A Matson, and Peter Vitousek (2011). *Principles of terrestrial ecosystem ecology*. Springer Science & Business Media.

Choisnel, EM, SV Jourdain, and CJ Jacquart (1995). "Climatological evaluation of some fluxes of the surface energy and soil water balances over France". In: *Annales Geophysicae*. Vol. 13. Copernicus GmbH, pp. 666–674. DOI: https://doi.org/10.1007/s00585-995-0666-y.

Coiffard, C, Bernard Gomez, and Frédéric Thévenard (2007). "Early Cretaceous angiosperm invasion of western Europe and major environmental changes". In: *Annals of Botany* 100.3, pp. 545–553. DOI: https://doi.org/10.1093/aob/mcm160.

De Boer, Hugo Jan et al. (2012). "A critical transition in leaf evolution facilitated the Cretaceous angiosperm revolution". In: *Nat. Commun.* 3.1, pp. 1–11. DOI: https://doi-org.insu.bib.cnrs.fr/10.1038.

Dow, Graham J and Dominique C Bergmann (2014). "Patterning and processes: how stomatal development defines physiological potential". In: *Curr. Opin. Plant Biol.* 21, pp. 67–74. DOI: https://doi.org/10.1016/j.pbi.2014.06.007.

Dow, Graham J, Dominique C Bergmann, and Joseph A Berry (2014). "An integrated model of stomatal development and leaf physiology". In: *New Phytol.* 201.4, pp. 1218–1226. DOI: 10.1111/nph.12608.

Ducharne, Agnes, Katia Laval, and Jan Polcher (1998). "Sensitivity of the hydrological cycle to the parametrization of soil hydrology in a GCM". In: *Climate dynamics* 14.5, pp. 307–327. DOI: https://doi.org/10.1007/s003820050226.

Dufresne, J-L et al. (2013). "Climate change projections using the IPSL-CM5 Earth System Model: from CMIP3 to CMIP5". In: *Clim. Dynam.* 40.9, pp. 2123–2165. DOI: https://doi.org/10.1007/s00382-012-1636-1.

Egea, Gregorio, Anne Verhoef, and Pier Luigi Vidale (2011). "Towards an improved and more flexible representation of water stress in coupled photosynthesis–stomatal conductance models". In: *Agricultural and Forest Meteorology* 151.10, pp. 1370–1384. DOI: https://doi.org/10.1016/j.agrformet.2011.05.019.

Farquhar, Graham D, S von von Caemmerer, and Joseph A Berry (1980). "A biochemical model of photosynthetic CO 2 assimilation in leaves of C 3 species". In: *Planta* 149.1, pp. 78–90. DOI: `https://doi.org/10.1007/BF00386231`.

Feild, Taylor S et al. (2011). "Fossil evidence for Cretaceous escalation in angiosperm leaf vein evolution". In: *P. Natl. Acad. Sci. USA* 108.20, pp. 8363–8366. DOI: `https://doi.org/10.1073/pnas.1014456108`.

Fiorin, Lucia, Timothy J Brodribb, and Tommaso Anfodillo (2016). "Transport efficiency through uniformity: organization of veins and stomata in angiosperm leaves". In: *New Phytologist* 209.1, pp. 216–227. DOI: `https://doi.org/10.1111/nph.13577`.

Gough, DO (1981). "Solar interior structure and luminosity variations". In: *Physics of solar variations*. Springer, pp. 21–34. DOI: `https://doi.org/10.1007/978-94-010-9633-1_4`.

Guimberteau, Matthieu et al. (2014). "Testing conceptual and physically based soil hydrology schemes against observations for the Amazon Basin". In: *Geosci. Model Dev.* 7.3, pp. 1115–1136. DOI: `https://doi.org/10.5194/gmd-7-1115-2014`.

Kattge, Jens et al. (2011). "TRY–a global database of plant traits". In: *Global change biology* 17.9, pp. 2905–2935. DOI: `https://doi-org.insu.bib.cnrs.fr/10.1111/j.1365-2486.2011.02451.x`.

Keenan, T et al. (2009). "Improved understanding of drought controls on seasonal variation in Mediterranean forest canopy CO 2 and water fluxes through combined in situ measurements and ecosystem modelling". In: *Biogeosciences* 6.8, pp. 1423–1444. DOI: `https://doi.org/10.5194/bg-6-1423-2009`.

Keenan, Trevor, Santi Sabate, and Carlos Gracia (2010). "Soil water stress and coupled photosynthesis–conductance models: Bridging the gap between conflicting reports on the relative roles of stomatal, mesophyll conductance and biochemical limitations to photosynthesis". In: *Agricultural and Forest Meteorology* 150.3, pp. 443–453. DOI: `https://doi.org/10.1016/j.agrformet.2010.01.008`.

Krinner, Gerhard et al. (2005). "A dynamic global vegetation model for studies of the coupled atmosphere-biosphere system". In: *Global Biogeochem. Cy.* 19.1. DOI: `https://doi.org/10.1029/2003GB002199`.

Lee, Jung-Eun and Kevin Boyce (2010). "Impact of the hydraulic capacity of plants on water and carbon fluxes in tropical South America". In: *J. Geophys. Res.-Atmos.* 115.D23. DOI: `https://doi.org/10.1029/2010JD014568`.

Pavlick, Ryan et al. (2013). "The Jena Diversity-Dynamic Global Vegetation Model (JeDi-DGVM): a diverse approach to representing terrestrial biogeography and biogeochemistry based on plant functional trade-offs". In: *Biogeosciences* 10.6, pp. 4137–4177. DOI: `https://doi.org/10.5194/bg-10-4137-2013`.

Ryder, James et al. (2016). "A multi-layer land surface energy budget model for implicit coupling with global atmospheric simulations". In: *Geoscientific Model Development* 9.1, pp. 223–245. DOI: `https://doi.org/10.5194/gmd-9-223-2016`.

Sakschewski, Boris et al. (2015). "Leaf and stem economics spectra drive diversity of functional plant traits in a dynamic global vegetation model". In: *Global Change Biology* 21.7, pp. 2711–2725. DOI: `https://doi.org/10.1111/gcb.12870`.

Scheiter, Simon, Liam Langan, and Steven I Higgins (2013). "Next-generation dynamic global vegetation models: learning from community ecology". In: *New Phytol.* 198.3, pp. 957–969. DOI: `https://doi.org/10.1111/nph.12210`.

Sepulchre, Pierre et al. (2020). "IPSL-CM5A2–an Earth system model designed for multi-millennial climate simulations". In: *Geosci. Model Dev.* 13.7, pp. 3011–3053. DOI: `https://doi.org/10.5194/gmd-13-3011-2020`.

Sewall, JO vd et al. (2007). "Climate model boundary conditions for four Cretaceous time slices". In: *Clim. Past* 3.4, pp. 647–657. DOI: https://doi.org/10.5194/cp-3-647-2007.

Tafasca, Salma, Agnès Ducharne, and Christian Valentin (2020). "Weak sensitivity of the terrestrial water budget to global soil texture maps in the ORCHIDEE land surface model". In: *Hydrology and Earth System Sciences* 24.7, pp. 3753–3774. DOI: https://doi.org/10.5194/hess-24-3753-2020.

Wheeler, Elisabeth A and Pieter Baas (2019). "Wood evolution: Baileyan trends and functional traits in the fossil record". In: *IAWA journal* 40.3, pp. 488–529. DOI: https://doi.org/10.1163/22941932-40190230.

Wilson, MF and A Henderson-Sellers (1985). "A global archive of land cover and soils data for use in general circulation climate models". In: *Journal of Climatology* 5.2, pp. 119–143. DOI: https://doi.org/10.1002/joc.3370050202.

Yin, X and PC Struik (2009). "C3 and C4 photosynthesis models: An overview from the perspective of crop modelling". In: *NJAS-Wagen. J. Life Sc.* 57.1, pp. 27–38. DOI: https://doi.org/10.1016/j.njas.2009.07.001.

Zaehle, Sönke and AD Friend (2010). "Carbon and nitrogen cycle dynamics in the O-CN land surface model: 1. Model description, site-scale evaluation, and sensitivity to parameter estimates". In: *Global Biogeochemical Cycles* 24.1. DOI: https://doi-org.insu.bib.cnrs.fr/10.1029/2009GB003521.

Zaehle, Sönke, AD Friend, et al. (2010). "Carbon and nitrogen cycle dynamics in the O-CN land surface model: 2. Role of the nitrogen cycle in the historical terrestrial carbon balance". In: *Global Biogeochemical Cycles* 24.1. DOI: https://doi-org.insu.bib.cnrs.fr/10.1029/2009GB003522.

Zhou, Shuangxi et al. (2014). "Short-term water stress impacts on stomatal, mesophyll and biochemical limitations to photosynthesis differ consistently among tree species from contrasting climates". In: *Tree physiology* 34.10, pp. 1035–1046. DOI: https://doi.org/10.1093/treephys/tpu072.

Zobler, Leonard (1986). "A world soil file grobal climate modeling". In: *NASA Tech. memo* 32.

---

## Author Response (AR1)

Dear editor,

Please find enclosed our revised manuscript and supplement. We have tried to improve our original manuscript as much as possible by taking into account all the recommendations. We sincerely hope that you are satisfied with our changes.

Concerning your minor comments on the discussion :

*Discussion with referee #1*
*1. Your reply to the comment on line 15:*
*I do not think "veinated" is a common word. Can you add a few words for explanation, may be in parentheses?*

We agree and we rephrased to "leaves with more and more veins".

*Discussion with referee #2*
*2. You reply to the comment on title:*
*I agree to modify the title, but is 'paleo-traits' the best word?*

As our purpose is to link fossil proxies to the modeled vegetation parameterization, we think that paleo-traits word seems appropriate. Nevertheless, we can propose an alternative if you think that "paleo-traits" is not clear enough : "The Cretaceous physiological adaptation of angiosperms to a declining pCO2: a modeling approach based on past leaf hydraulic and photosynthetic capacities"

*3. You reply to the comment on line 110:*
*As long as I know, the third term of biochemical photosynthesis model is related to triose phosphate limitation (for substrate transportation). A recent study (Rogers et al., 2021) assessed critically this term, being in line with your decision. Just for your information.*

Thank you for this comment, this study is very interesting.

*4. Your reply on Figure 3:*
*In the new Figure 3, 'bare soil' is shown by dark green. I think it's confusing with other forest biomes. Also, 'C3 grass' may be '(Cretaceous) savannas', according to your reply to the comment on line 222.*

Sorry for the confusing colors between bare soil and forest biomes. Now you can see that we have drawn bare soil in very light yellow.

Figure 3 shows the dominant ORCHIDEE PFT but not biomes as Figure 6 in Sewall et al., (2007) already shows biomes distribution with savannas. To make it clearer we add sentences lines 201-202 in the manuscript : "Figure 3 shows only the dominant PFT, but several PFTs can be present in a pixel. For instance, regions indicated as C3 grass are savannas with a fraction of trees (Table S1)" and we explain our choices in the supplement after Table S1."

Thank you again for your consideration.

Sincerely yours,

Julia Bres and co-authors.